# SCORE: Similarity-Aware Contextual Overlap-Redundancy Eviction for Efficient KV Cache Compression in LLMs

## Abstract

Recent advances in large language models (LLMs) have unlocked remarkable long-context capabilities, enabling breakthroughs across diverse NLP tasks. However, despite architectural progress and compression techniques such as quantization, the key-value (KV) cache remains a critical memory bottleneck during inference. Prior work has explored cache optimization via eviction strategies, yet most rely on heuristic or single-axis importance metrics, neglecting the nuanced and dynamic interplay between layers and attention heads. In this paper, we propose *SCORE* (Similarity-aware Contextual Overlap-Redundancy Eviction), a novel framework that introduces a distance-based multi-level similarity metric to quantify and eliminate structural redundancy within the KV cache. By dynamically reallocating cache budgets across layers and heads and employing a redundancy-aware greedy token selection mechanism, *SCORE* preserves semantic diversity while minimizing memory overhead. Extensive experiments on long-context benchmarks such as LongBench and NeedleBench show that *SCORE* retains 95% of full KV cache performance using only 1.5% of the cache, consistently outperforming state-of-the-art baselines under strict memory constraints. These results underscore the value of fine-grained, context-aware cache management for scalable and efficient long-context inference in LLMs.

## 1 Introduction

Large language models (LLMs) have shown exceptional long-context understanding, achieving state-of-the-art performance across a wide range of natural language processing (NLP) tasks, including multi-turn dialogue, document summarization, and information retrieval Zhao et al. (2023). Recent models, such as GPT-4 Achiam et al. (2023), Claude 3.5 Anthropic (2024), LLaMA 3.1 Grattafiori et al. (2024), and Mistral Jiang et al. (2023), demonstrate significantly improved long-context capabilities, with some supporting up to one million tokens Anthropic (2024), enabling the stable processing of substantially extended input sequences. Moreover, recent advances—accompanied by various compression techniques such as quantization Kim

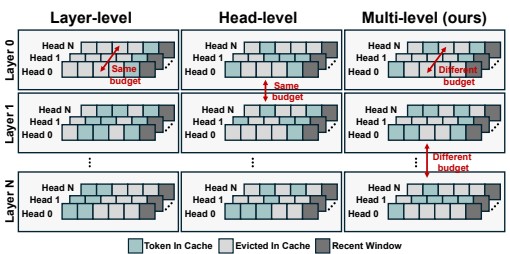

Figure 1: KV cache eviction strategies: (a) Layer-wise allocation assigns distinct budgets across layers; (b) Head-wise allocation enables finer-grained control; (c) SCORE combines both to reduce redundancy and optimize cache usage.

et al. (2024); Choi & Kim (2025)—have further accelerated efforts to deploy LLMs in resource-constrained environments, including on-device settings Kwon et al. (2023); Xu et al. (2024); Li et al. (2024b). However, the key-value (KV) cache remains a major memory bottleneck for long-context processing, significantly limiting efficiency.

Accordingly, recent research has focused on optimizing the KV cache without altering model architecture Dao (2023); Acharya et al. (2024). A key direction is the eviction of low-importance KV pairs to reduce memory usage Ge et al. (2023); Zhang et al. (2023); Xiao et al. (2023), typically guided by

attention scores or heuristic rules. However, as shown in Dong et al. (2021); Zhang et al. (2022), the contribution of layers and attention heads to text generation is highly uneven and context-dependent. Efficient cache utilization thus requires fine-grained, dynamic strategies that account for this variability. To address such non-uniformity, prior work has explored both layer-level and head-level budgeting. For example, PyramidKV Cai et al. (2024) allocates cache budgets based on information density across layers, whereas Cake Qin et al. (2025) employs a cascading mechanism to adjust budgets during prefill, as illustrated in Figure 1(a). More fine-grained methods like SnapKV Li et al. (2024c) and AdaKV Feng et al. (2024) estimate head-level importance and reallocate budgets accordingly, as shown in Figure 1(b). However, these methods rely primarily on simple statistics—such as entropy or variance—without leveraging richer signals (*L1*). Moreover, they fail to jointly capture redundancy across layer–head interactions, which further limits their effectiveness (*L2*). In addition, their simplistic budget allocation often leads to limited token diversity, resulting in poor coverage and making it difficult to capture the overall context (*L3*).

To mitigate these limitations, we introduce **S**imilarity-aware **C**ontextual **O**verlap-**R**edundancy **E**viction (*SCORE*), a cache management framework that quantifies representational redundancy across layers and heads using a distance-based multi-level similarity metric, as illustrated in Figure 1(c). *SCORE* enables the removal of semantically redundant tokens while preserving contextual diversity through dynamic budget reallocation and selective cache retention. This design captures representational divergence and contextual progression more effectively than prior heuristics-based methods, enhancing redundancy-aware information preservation.

The main contributions of *SCORE* are as follows: (i) *SCORE* is the first to introduce a distance-based metric to precisely measure and eliminate redundancy within the KV cache (for *L1*). (ii) **Redundancy-aware multi-level metric.** To capture hierarchical information flow and representational diversity, we introduce multi-level similarity metrics that quantify redundancy across layers and heads in the KV cache (for *L2*). (iii) **Hierarchical budget allocation.** The proposed *SCORE* framework dynamically reallocates cache budgets across layers and heads based on redundancy scores, prioritizing informative and non-redundant tokens under constrained memory. (iv) **Greedy token selection.** Our redundancy-aware, greedy token selection algorithm maximizes information diversity in the cache by accounting for similarity (for *L3*).

## 2 BACKGROUNDS

### 2.1 INFERENCE OPERATIONS WITH KV CACHE

Inference in transformer-based LLMs typically consists of two stages: a prefill stage that encodes the input sequence, and a decode stage that sequentially generates output tokens.

**Prefill stage:** Given an input prompt tensor $X \in \mathbb{R}^{S \times D}$, where $S$ denotes the sequence length and $D$ the hidden dimension, the key and value representations are computed as follows:

$$Q = XW_Q, \ K = XW_K, \ V = XW_V, \tag{1}$$

where, $W_Q, W_K, W_V \in \mathbb{R}^{D \times D}$ are the learnable projection matrices that map the input sequence to query, key, and value representations, respectively. The resulting key and value tensors are then stored in the KV cache, where they are reused during the subsequent decode stage to avoid redundant computation.

**Decode stage:** During decoding, for each newly generated token $x_i \in \mathbb{R}^{1 \times D}$ at time step $i$, the corresponding key and value are computed, while previous information is retrieved from the KV cache. The new key and value are then appended to the existing cache along the sequence dimension via concatenation, as follows:

$$K \leftarrow K \cup x_i W_K, \ V \leftarrow V \cup x_i W_V, \tag{2}$$

Then, the current query $q_i = x_i W_Q$ computes attention weights via a scaled dot-product with the full $K$, and aggregates $V$ accordingly to produce the output $x_{i,\text{out}}$ as follows:

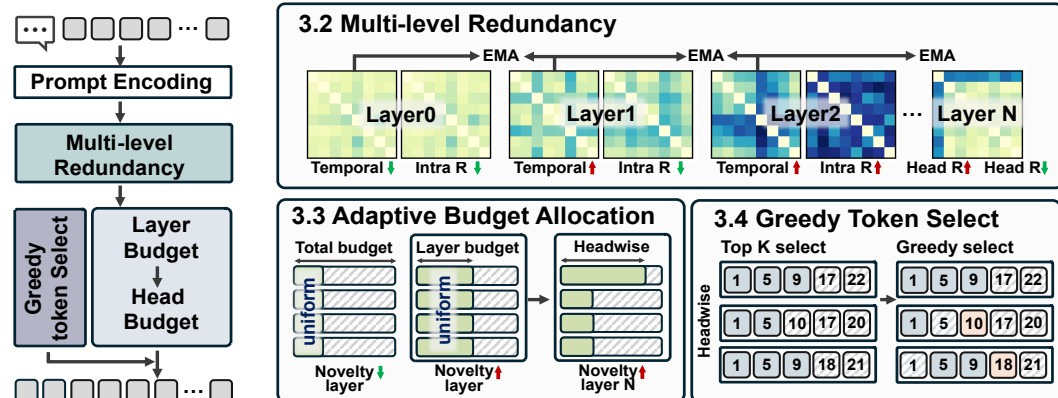

Figure 2: An overview of SCORE. The model computes redundancy scores across layers and heads, which guide a dynamic budget allocation strategy and inform redundancy-aware token selection to achieve diverse representation. For each layer: left shows temporal redundancy with the previous layer; right shows intra-layer head similarity.

$$x_{i,\text{out}} = \text{Softmax}\left(\frac{q_i K^\top}{\sqrt{D}}\right) V \tag{3}$$

As decoding progresses, the size of the KV cache increases linearly (*e.g.*, 62.5GB in Llama3-8B-Instruct Shi et al. (2024)), which becomes a major bottleneck in terms of memory usage and latency. Efficient cache compression and management are therefore essential for long-context processing.

## 2.2 KV CACHE EVICTION

H2O Zhang et al. (2023) improves efficiency by focusing on important tokens (heavy hitters), while ROCO Ren & Zhu (2024) and Scissorhands Liu et al. (2023) similarly retain key tokens based on attention scores. StreamingLLM Xiao et al. (2023) and LM-Infinite Han et al. (2023) prioritize nearby tokens relevant to generation, but uniform eviction often leads to information loss. To address this, dynamic budget allocation methods have emerged. PyramidKV Cai et al. (2024) employs a pyramidal attention pattern, allocating more cache to lower layers while summarizing higher-layer information. CAKE Qin et al. (2025) analyzes attention dispersion and temporal shifts to reassign cache budgets in a cascading manner. However, both rely solely on layer-level budgeting. To enable finer granularity, SnapKV Li et al. (2024c) clusters attention distributions at the head level, and HeadKV Fu et al. (2024) introduces a theoretical allocation scheme to minimize post-eviction degradation. AdaKV Feng et al. (2024) estimates head-level importance based on retrieval and reasoning contributions. While these methods achieve budgeting at either the layer or head level (*L2*), they fail to jointly consider both dimensions and rely heavily on attention scores (*L1*), which limits their ability to capture semantic redundancy. Moreover, even with well-determined budgets, token-level diversity is overlooked (*L3*), leading to overlapping selections and limited coverage. To overcome these limitations, we propose *SCORE*, a unified framework that quantifies redundancy via distance-based similarity and integrates it into both budget allocation and token eviction.

## 3 PROPOSED METHOD

### 3.1 OVERALL ARCHITECTURE OF SCORE

As illustrated in Figure 2, *SCORE* is a redundancy-aware KV cache management framework. It consists of three core components: a multi-level redundancy scoring module (Section 3.2) that quantifies representational redundancy across layers and heads; a dynamic budget allocation strategy (Section 3.3) that selectively retains KV entries based on their relative importance; and a token selection mechanism (Section 3.4) that prioritizes diverse and informative tokens for cache retention. Algorithm 1 summarizes the end-to-end pipeline of the proposed *SCORE* framework.

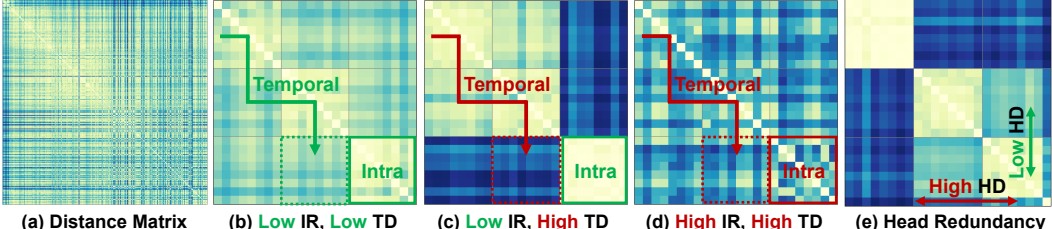

Figure 3: Head-wise cosine distance matrices. From left to right: (a) the full similarity matrix $\mathbf{D}$ across all heads and layers; (b)–(d) three examples of 3×3 layer-wise similarity matrices, each showing different levels of intra-layer redundancy and inter-layer diversity; (e) an illustrative example highlighting head-level redundancy. Brighter colors indicate higher similarity.

## 3.2 MULTI-LEVEL REDUNDANCY SCORING

According to prior studies Michel et al. (2019); Dalvi et al. (2020), many layers and attention heads in transformer models often learn functionally similar representations, leading to structural redundancy. To quantitatively assess such redundancy and diversity, we propose multi-level evaluation metrics based on inter-head similarity and mitigate *L1*, *L2*. Specifically, we sample $k$ token positions from each head and compute the average pairwise distance of their attention scores. Given a distance function $\delta(\cdot, \cdot)$ between two attention scores, the distance between head $i$ and head $j$ is defined as:

$$D_{(i,j)} = \frac{1}{k^2} \sum_{s=0}^{k-1} \sum_{t=0}^{k-1} \delta(\mathbf{a}_i^{(s)}, \mathbf{a}_j^{(t)}), \tag{4}$$

where $\mathbf{a}_i^{(s)} \in \mathbb{R}^n$ denotes the attention score vector of head $i$ at the $s$-th sampled token for indices $i, j \in \{0, 1, \ldots, (L \cdot H - 1)\}$. Figure 3(a) visualizes the distance matrix $D$, which contains pairwise distances between attention heads across all layers. It intuitively reveals that redundant representations are often concentrated in specific layers or heads, serving as the foundation for the hierarchical analysis metrics proposed in the subsequent sections.

**Intra-layer redundancy.** To analyze the extent of redundant representations within each layer, we define the intra-layer redundancy ($\mathcal{IR}$) of layer $l$ as the average similarity across its attention heads, computed by evaluating a similarity measure $D$ over all pairs of heads $(h, h')$ within the same layer:

$$\mathcal{IR}_l = \frac{1}{H^2} \sum_{h,h'=0}^{H-1} D_{lH+h, \, lH+h'}, \tag{5}$$

Here, lower $\mathcal{IR}_l$ indicates higher redundancy due to increased similarity across heads, whereas higher $\mathcal{IR}_l$ suggests greater functional diversity.

**Temporal deviation.** To quantify the degree of novel information introduced relative to the preceding layer, we introduce the temporal deviation ($\mathcal{TD}$) metric. It tracks changes across layers by continuously comparing the similarity between adjacent outputs against an exponential moving average (EMA). For layer $l$, $\mathcal{TD}_l$ is defined as follows:

$$\mathcal{TD}_l = |\mu_l - \mathrm{E}_{l-1}|, \tag{6}$$

where $\mu_l$ denotes the average similarity between layer $l$ and its previous layer, computed in a manner similar to Eq. 5, but based on the distance matrix $D$ between two distinct layers rather than within a single layer. For the first layer, we set $\mathcal{TD}_0 = \mathcal{IR}_0$. $\mathrm{E}_{l-1}$ represents the accumulated EMA-based deviation up to layer $l - 1$, and is updated as follows:

$$\mathrm{E}_l = \gamma \cdot \mathrm{E}_{l-1} + (1 - \gamma) \cdot \mu_l, \tag{7}$$

where $\gamma$ controls responsiveness to temporal shifts in layer-wise representations. Higher $\mathcal{TD}_l$ indicates that layer $l$ produces more novel and distinctive outputs compared to the accumulated patterns of preceding layers. Figure 2 (specifically, inside the box corresponding 3.2) illustrates this by comparing the left temporal matrices across layers, where notable increases in deviation reveal points of significant representational change. Figures 3(b)-(d) present varied examples with different levels of redundancy and temporal variation, clarifying the hierarchical structure and functional differentiation. Additional visualizations are provided in Appendix D.2.

**Head-level distinctness.** Beyond analyzing redundancy at the layer level, we introduce the head-level distinctness ($\mathcal{HD}$) metric to quantitatively assess the uniqueness of individual attention heads. This metric quantifies how distinct a head's attention pattern is compared to others in the same and preceding layers. An illustrative example is shown in Figure 3(e). For a head $h$ in layer $l$, $\mathcal{HD}$ is defined as follows:

$$\mathcal{HD}_{l,h} = \frac{1}{2H} \sum_{i=0}^{1} \sum_{h'=0}^{H-1} D_{lH+h,\,(l-i)H+h'} \quad (8)$$

Here, higher $\mathcal{HD}_{l,h}$ indicates the head is more distinguishable from its neighbors, suggesting greater likelihood of fulfilling a unique functional role. The metric complements layer-level averages by enabling fine-grained evaluation of diversity and redundancy at the head level.

---

**Algorithm 1** Multi-level budget allocation

**Input:** accumulate attention score $A_l \in \mathbb{R}^{H \times S}$, number of layers L, number of heads H, Total cache budget $B_{total}$

**Output:** Retained cache set $C = \{K_{l,h}^{(m)}, V_{l,h}^{(m)}\}$

  *// Layer-wise budget allocation. Cascading strategy adapted from CAKE*
1: **for** stage $m = 0$ to $L - 1$ **do**
2: $\quad C^{(m)} \leftarrow C^{(m-1)} \cup \{K^{(m)}, V^{(m)}\}$
3: $\quad$ Compute distance matrix $D \in \mathbb{R}^{H \times 2H}$ using Eq. 4
4: $\quad \mathcal{IR}, \mathcal{TD}$ are computed by slicing $D$ (Eq. 5- 7)
5: $\quad \mathcal{IR}^{(m)} \leftarrow \mathcal{IR}^{(m-1)} \cup \mathcal{IR}_m,$
   $\quad \mathcal{TD}^{(m)} \leftarrow \mathcal{TD}^{(m-1)} \cup \mathcal{TD}_m$
6: $\quad B^{(m)} \leftarrow \{B_l^{(m)} \mid l \in [0,m]\}$, where each $B_l^{(m)}$ is computed according to Eq. 10
   *// Head-wise budget allocation and evict*
7: $\quad$ Extract attention matrix $A_m \in \mathbb{R}^{H \times S}$
8: $\quad \Omega_m \leftarrow$ select top-$B_m^{(m)}$ token with attention
9: $\quad$ Compute $T_{m,h}$ by counting token-to-head assignments using Eq. 11
10: $\quad$ **for** head $h = 0$ to $H - 1$ **do**
11: $\quad\quad$ Compute head distinctness $HD_{m,h}$ Eq. 8
12: $\quad\quad$ Allocate head-wise budget $B_{m,h}^{(m)}$ using $T_{m,h}$ and $\mathcal{HD}_{m,h}$ as in Eq. 12
    *// Greedy token selection*
13: $\quad\quad$ Penalized attention score $\widetilde{A}_m$ using Eq. 13
14: $\quad\quad I_{m,h}^{(m)} \leftarrow \text{TopK}\big(\widetilde{A}_m[h],\, B_{m,h}^{(m)}\big)$
15: $\quad\quad \hat{K}_{m,h}^{(m)}, \hat{V}_{m,h}^{(m)} \leftarrow K_{m,h}^{(m)}\big[I_{m,h}^{(m)}\big], V_{m,h}^{(m)}\big[I_{m,h}^{(m)}\big]$
16: $\quad\quad C^{(m)}[m][h] \leftarrow \{\hat{K}_{m,h}^{(m)}, \hat{V}_{m,h}^{(m)}\}$
17: $\quad$ **end for**
18: **end for**
19: **return** $C = \{K_{l,h}^{(L-1)}, V_{l,h}^{(L-1)}\}$

---

### 3.3 Hierarchical Budget Allocation

**Layer budget allocation.** We build on the cascading strategy from Qin et al. (2025) and introduce adaptive budget reallocation driven by the importance of hierarchical representations. At each stage, only interactions between active layers and their immediate predecessors are considered. Accordingly, the pairwise distance matrix $D$ is computed only between adjacent layers. Cache budgets are dynamically assigned based on a novelty score $\mathcal{N}_l$, which integrates inter-layer redundancy ($\mathcal{IR}_l$) and temporal dynamics ($\mathcal{TD}_l$) as follows:

$$\mathcal{N}_l = \lambda_1 \cdot \mathcal{IR}_l + \lambda_2 \cdot \mathcal{TD}_l, \quad (9)$$

where $\lambda_1, \lambda_2 \in \mathbb{R} \geq 0$ are weighting factors that control the relative importance of each term. The novelty score $\mathcal{N}l$ quantifies the diversity and informational distinctiveness of layer $l$, and guides the allocation of the total computational budget $B_{\text{total}}$:

$$B_l^{(m)} = \frac{\mathcal{N}_l}{\sum_{k=0}^{L-1} \mathcal{N}_k} \cdot B_{\text{total}}, B_l^{(m)} < B_l^{(m-1)}, \quad (10)$$

At cascading stage $m$, the cache budget for layer $l$ is assigned in proportion to its novelty score, enabling adaptive allocation based on importance (Algorithm 1, lines 1–6).

**Head budget allocaction.** As shown in Figure 2, cache budgets are first allocated at the layer level and then uniformly distributed across heads. To account for variations in novelty and importance across heads, we propose a fine-grained, head-wise allocation strategy. Given the accumulated attention score matrix $\mathbf{A}_l \in \mathbb{R}^{H \times S}$ for layer $l$, we select the top-$B_l$ scores globally. Let $\Omega_l$ denote the corresponding set of important tokens. For each token $s \in \Omega_l$, we identify its source head and compute head-wise contributions accordingly:

$$T_{l,h} = |\{(h', s) \in \Omega_l \mid h' = h\}|, \tag{11}$$

The resulting vector $\mathbf{T}_l = [T_{l,1}, T_{l,2}, \ldots, T_{l,H}] \in \mathbb{R}^H$ therefore serves as an explicit representation of the relative importance of each head, capturing how frequently individual heads contribute to the set of top-ranked tokens. This is then combined with the head redundancy measure $\mathcal{HD}_{l,h}$, to determine the head-wise budget $B_{l,h}$ as follows:

$$B_{l,h} = B_l \cdot \left( \frac{HD_{l,h}}{\sum_{h'=1}^{H} HD_{l,h'}} \right) \cdot T_{l,h} \tag{12}$$

This strategy refines layer-level cache allocation by jointly considering head-wise relevance and redundancy, effectively addressing the **L2** and preserving distinctive information. It serves as a key mechanism for improving eviction performance in LLMs (Algorithm 1, lines 7–17).

### 3.4 GREEDY TOKEN SELECTION

The multi-level budget allocation strategy improves cache efficiency by quantitatively assessing redundancy and contribution at both the layer and head levels. However, despite this fine-grained allocation, redundant heads often select overlapping tokens, concentrating attention on specific positions and leading to coverage bias. The 'Top-K select' example in Figure 2 illustrates this overlooked limitation.

To mitigate this issue, we propose a redundancy-aware soft selection mechanism that promotes diversity across attention heads during token selection. The method lowers the selection priority of tokens redundantly chosen by multiple heads, with the penalty strength determined by the degree of head-wise diversity within the layer. By discouraging redundant selection, this approach alleviates head-level concentration and fosters more diverse and informative representations. The resulting penalized attention score is defined as follows:

$$\widetilde{A}_l[h, s] = A_l[h, s] \cdot \exp\left( -\frac{\alpha}{\mathcal{HD}_{l,h}} \cdot r_s \right) \tag{13}$$

where $\alpha > 0$ is a hyperparameter that controls the attenuation strength of the penalty, and $r_s$ denotes the number of times token $s$ has been selected across multiple heads. Higher redundancy leads to stronger penalties, encouraging greater dispersion in head-wise token selection. This method can be seamlessly integrated into existing token selection pipelines with little additional overhead, while effectively enhancing informational diversity and addressing the **L3** by ensuring that different heads contribute complementary rather than redundant evidence.

## 4 EXPERIMENTS AND ANALYSIS

### 4.1 EXPERIMENTAL SETUP

**Baseline Models.** We evaluate representative open-source LLMs with context lengths from 4K to 128K tokens, including two multi-head attention models: Llama2-Chat Touvron et al. (2023) (7B and 13B), and two grouped-query attention models: Llama3-8B-Instruct Grattafiori et al. (2024) and Mistral-7B-Instruct-v0.3 Jiang et al. (2023). To evaluate memory allocation strategies under constrained cache budgets, we adopt CAKE Qin et al. (2025), which dynamically reallocates cache in a cascading manner during prefilling to retain informative tokens.

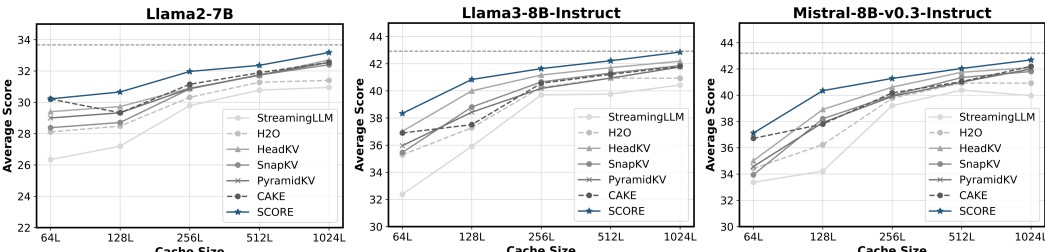

Figure 4: Mean performance across 16 LongBench datasets for varying KV cache sizes. The dashed line indicates the performance with the full KV cache.

Table 1: Performance comparison over 16 datasets from LongBench. Results are measured with a cache size of 128. The best score is marked in **bold**, and the second best is marked with underline.

| | Method | Single-Document QA | | | Multi-Document QA | | | Summarization | | | Few-shot Learning | | | Synthetic | | Code | |
|---|---|---|---|---|---|---|---|---|---|---|---|---|---|---|---|---|---|
| | | NtvQA | Qasper | MF-en | HotpotQA | 2WikiMQA | Musique | GovReport | QMSum | MultiNews | TREC | TriviaQA | SAMSum | PCount | PR-en | Lcc | RB-P |
| **Llama2-7B** | Full KV | 18.39 | 21.13 | 35.54 | 31.35 | 25.61 | 10.64 | 25.57 | 22.18 | 26.26 | 65.5 | 89.41 | 41.03 | 6.00 | 8.50 | 58.67 | 53.00 |
| | StreamingLLM | 11.27 | 15.93 | 28.45 | 24.77 | 20.92 | 6.46 | 15.42 | 16.8 | 18.97 | 60.55 | 79.42 | 35.32 | 3.00 | 2.00 | 50.82 | 45.22 |
| | H2O | 11.82 | 17.75 | 29.70 | 24.98 | 21.95 | 8.33 | 16.98 | 18.49 | 20.49 | 62.50 | 80.79 | 36.78 | 3.50 | 3.00 | 51.95 | 46.68 |
| | HeadKV | 14.86 | 19.25 | 28.40 | 27.39 | 24.30 | 8.38 | 18.29 | 20.01 | 21.10 | 63.50 | 81.99 | 37.92 | 4.00 | 4.50 | 53.54 | 48.34 |
| | SnapKV | 14.12 | 18.50 | 27.25 | 28.45 | 21.83 | 7.94 | 17.23 | **20.04** | 20.30 | 58.00 | 81.85 | 37.43 | 4.00 | 4.00 | 52.05 | 46.41 |
| | PyramidKV | 15.20 | 18.70 | 29.02 | 29.72 | 23.57 | 7.95 | 17.87 | 20.01 | 20.32 | 60.50 | 82.59 | 36.71 | 4.00 | 4.00 | 52.58 | 46.68 |
| | CAKE | 15.62 | 19.01 | 32.10 | 29.46 | 24.78 | 9.48 | 18.75 | 20.01 | **22.52** | 63.50 | 82.95 | **38.75** | 4.50 | 6.00 | 54.59 | 48.66 |
| | SCORE(ours) | **16.90** | **19.84** | 35.49 | **30.52** | 24.88 | 10.55 | 19.30 | 20.02 | 22.32 | **64.10** | **84.00** | 38.86 | 5.50 | 7.00 | 56.45 | 49.63 |
| **Llama2-13B** | Full KV | 19.15 | 26.38 | 36.77 | 36.57 | 33.93 | 14.32 | 25.89 | 20.33 | 26.06 | 65.00 | 87.70 | 35.57 | 6.50 | 10.50 | 51.26 | 53.40 |
| | StreamingLLM | 12.02 | 8.21 | 22.33 | 9.37 | 7.49 | 2.63 | 18.40 | 18.81 | 18.93 | 61.05 | 84.35 | 40.00 | 2.00 | 13.05 | 35.02 | 36.75 |
| | H2O | 13.29 | 11.44 | 24.15 | 12.84 | 10.38 | 3.65 | 20.75 | 20.21 | 20.99 | 64.50 | 86.77 | 40.41 | **3.60** | 15.25 | 38.13 | 39.42 |
| | HeadKV | 13.20 | 13.04 | 25.27 | 12.09 | 11.00 | 2.73 | 20.72 | 20.30 | 22.20 | 68.00 | 86.00 | 39.55 | 3.07 | 14.75 | 41.36 | 41.06 |
| | SnapKV | 12.61 | 11.91 | 23.78 | 13.71 | 9.98 | 3.87 | 19.97 | 19.96 | 21.40 | 63.50 | 86.75 | 39.89 | 3.00 | **16.75** | 40.51 | 38.57 |
| | PyramidKV | 13.64 | 11.50 | 26.04 | 14.01 | 10.60 | **5.40** | 20.32 | 19.63 | 21.65 | 64.50 | 86.25 | 39.54 | 3.50 | 16.25 | 41.08 | 39.69 |
| | CAKE | 13.64 | 11.87 | 25.41 | 11.80 | 10.49 | 4.50 | 20.94 | 20.30 | 21.77 | 65.50 | 86.86 | 42.48 | 3.50 | 15.75 | 39.80 | 38.98 |
| | SCORE(ours) | 14.06 | 14.64 | 27.87 | 11.71 | 11.14 | 5.16 | **21.98** | **20.68** | 24.06 | 69.65 | **87.96** | 41.82 | **3.60** | 16.50 | **44.96** | **44.82** |
| **Llama3-8B-Instruct** | Full KV | 25.56 | 39.43 | 45.23 | 45.37 | 35.65 | 21.63 | 28.63 | 23.35 | 26.81 | 74.00 | 90.48 | 42.52 | 4.80 | 69.25 | 56.97 | 52.42 |
| | StreamingLLM | 20.58 | 28.66 | 25.17 | 37.45 | 18.72 | 18.87 | 19.01 | 20.17 | 18.81 | 62.02 | 88.57 | 38.96 | 3.51 | 67.79 | 55.42 | 50.86 |
| | H2O | 21.90 | 30.11 | 26.55 | 38.95 | 20.01 | 20.29 | 20.37 | 21.58 | 20.18 | 63.50 | 89.96 | **40.39** | 4.40 | 69.25 | 56.83 | 52.30 |
| | HeadKV | 22.47 | 30.13 | **40.38** | **44.90** | **31.06** | **21.10** | 20.70 | 22.31 | 21.91 | 71.00 | **90.82** | 39.62 | 4.35 | **69.50** | 57.65 | 52.65 |
| | SnapKV | 22.17 | 28.96 | 36.29 | 42.10 | 29.25 | 19.78 | 20.11 | 22.56 | 21.46 | 66.00 | 89.72 | 38.89 | 4.50 | 69.00 | 57.24 | **52.97** |
| | PyramidKV | 22.10 | 26.94 | 36.86 | 40.38 | 29.42 | 16.34 | 20.34 | **22.70** | 21.99 | 67.00 | 89.35 | 39.77 | 4.50 | 69.00 | 56.55 | 51.67 |
| | CAKE | 22.10 | 32.19 | 34.52 | 39.06 | 30.45 | 20.72 | 20.40 | 21.85 | 20.98 | 46.00 | 89.64 | 39.74 | 4.50 | 69.00 | 56.46 | 52.14 |
| | SCORE(ours) | **24.65** | **33.31** | 40.22 | 45.27 | 34.29 | 21.45 | 21.63 | 22.36 | **22.93** | 72.30 | 90.66 | 40.20 | **5.00** | **69.50** | 57.03 | 52.76 |
| **Mistral-7B-Instruction-v0.3** | Full KV | 29.53 | 41.58 | 53.13 | 49.22 | 39.51 | 28.58 | 34.68 | 26.42 | 27.82 | 80.50 | 92.14 | 47.44 | 5.50 | 98.00 | 58.45 | 59.54 |
| | StreamingLLM | 22.83 | 25.80 | 41.54 | 39.11 | 23.73 | 15.44 | 18.04 | 16.34 | 15.86 | 45.55 | 83.82 | 37.55 | 3.00 | 75.00 | 43.54 | 40.46 |
| | H2O | 23.29 | 27.23 | 42.08 | 40.68 | 24.86 | 16.48 | 18.75 | 17.65 | 17.07 | 56.50 | 85.03 | 38.88 | 4.50 | 80.00 | 44.53 | 42.44 |
| | HeadKV | 26.04 | 30.25 | 47.93 | 43.17 | **32.13** | 22.63 | **20.73** | 18.24 | 18.79 | 65.50 | 85.29 | 40.69 | 5.00 | 76.50 | 44.93 | 44.95 |
| | SnapKV | 25.72 | 28.56 | 46.34 | 43.52 | 29.10 | 20.86 | 19.51 | 18.09 | 18.49 | 64.50 | 84.81 | 40.20 | **5.50** | 78.50 | 43.99 | 43.68 |
| | PyramidKV | 25.50 | 27.47 | 46.23 | 44.02 | 30.05 | 21.10 | 20.08 | 18.15 | 18.10 | 63.50 | 85.02 | 40.37 | 4.50 | 78.50 | 43.52 | 40.73 |
| | CAKE | **26.17** | 27.26 | 43.73 | 41.30 | 25.89 | 17.30 | 20.22 | 18.01 | 17.65 | 57.50 | 85.53 | 39.11 | **5.00** | **82.00** | 45.89 | 43.39 |
| | SCORE(ours) | 25.95 | **32.65** | 50.54 | **44.44** | 31.91 | 23.89 | 20.73 | 19.89 | 19.95 | 69.00 | 86.33 | 41.28 | 4.50 | 81.50 | 46.54 | 46.61 |

**Tasks.** We evaluate model performance under compressed KV cache settings using LongBench Bai et al. (2023), which covers a range of long-context tasks. For fine-grained retrieval and long-range reasoning, we use NeedleBench Li et al. (2024a) and Reasoning-in-a-Haystack Kuratov et al. (2024), respectively. We also include Longbench V2 Bai et al. (2024) and InfiniteBench Zhang et al. (2024) for extreme long-context settings.

**Implementation.** All experiments were conducted on an NVIDIA A100 80GB GPU with cache budgets ranging from 64L to 2048L. To ensure fair comparison, all methods were evaluated under identical conditions and cache capacities. Hyperparameter details are provided in Appendix A.

## 4.2 MAIN RESULTS

**Evaluation on Longbench.** We evaluate *SCORE* on the LongBench benchmark, which includes 16 long-context tasks. All experiments are conducted under identical conditions for fair comparison. As shown in Figure 4, *SCORE* consistently outperforms baseline methods across varying KV cache budgets, with a significant advantage in low-cache settings ($B_{total} \leq 128L$) due to its ability to preserve contextual diversity by selectively retaining salient tokens. Table 1 reports results in low-cache scenarios, where *SCORE* outperforms existing methods on most tasks. On LLaMA3-8B, it achieves an average score of 40.85—surpassing HeadKV by +0.82—while retaining 95.1% of full-cache performance. Full results across cache sizes are available in Appendix E.

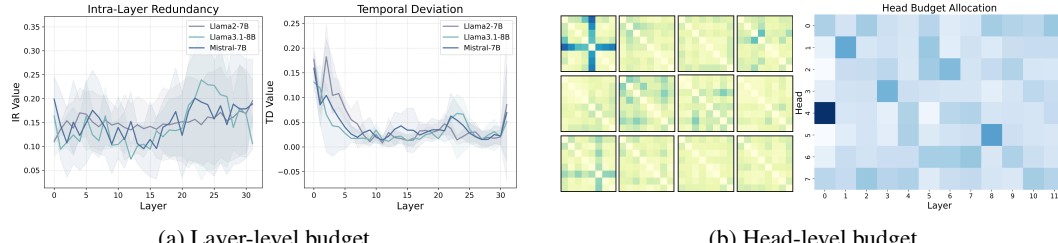

(a) Layer-level budget          (b) Head-level budget

Figure 5: Redundancy metrics used for budget allocation. (a) Layer-wise distributions of $\mathcal{IR}$ and $\mathcal{TD}$ averaged over the entire LongBench dataset; head-wise redundancy for the first 12 layers on the NarrativeQA and corresponding head-wise budget allocation.

**Evaluation on LongBench V2.** We further evaluate *SCORE* under realistic scenarios using LongBench-v2, a long-context benchmark designed to overcome the limitations of prior datasets that relied heavily on synthetic data or extraction-style tasks. It covers six categories and twenty sub-tasks—including QA, in-context learning, dialogue, code, and structured data reasoning—with context lengths ranging from 8K up to 2M words. Table 2 reports the detailed results on LLaMA3-8B-Instruct. Across both cache sizes (128 and 1024), *SCORE* consistently outperforms competitive baselines, indicating that the proposed method provides benefits regardless of the memory budget available. In the

Table 2: Performance comparison on Llama3-8B-Instruct over LongBench V2. Results are measured with cache sizes of 128 and 1024.

| Method | Easy | Hard | Short | Medium | Long | Avg. |
|---|---|---|---|---|---|---|
| Full KV | 31.25 | 25.08 | 34.44 | 24.19 | 22.22 | 27.44 |
| Llama3-8B-Instruct, Cache size=128 | | | | | | |
| CAKE | 29.27 | 21.13 | **32.98** | 21.22 | 20.18 | 24.96 |
| HeadKV | 29.53 | 22.74 | 32.25 | 22.97 | 20.10 | 25.52 |
| SCORE | **29.94** | **23.11** | 32.78 | **23.73** | **20.33** | **25.98** |
| Llama3-8B-Instruct, Cache size=1024 | | | | | | |
| CAKE | 29.50 | 21.31 | 33.25 | 22.61 | 21.06 | 25.55 |
| HeadKV | 30.22 | 23.97 | 33.24 | 23.23 | 20.14 | 26.16 |
| SCORE | **30.25** | **24.08** | **33.89** | **24.19** | **21.45** | **26.77** |

extreme compression setting with cache size 128, where most baselines struggle to maintain stable accuracy, *SCORE* achieves an average score of 25.98, surpassing HeadKV and CAKE by +0.46 and +1.02, respectively. Importantly, this performance gap is not limited to a particular type of task but persists across difficulty levels and input lengths. In particular, *SCORE* demonstrates clear advantages in the Hard and Medium scenarios, where long-range reasoning and multi-step comprehension are crucial. These results highlight that the method is capable of preserving critical information under severe compression, achieving stable performance even in extreme long-context conditions, which suggests strong potential for deployment in practical large-scale applications.

## 4.3 JUSTIFYING REDUNDANCY-AWARE MODELING

**Redundancy Observations**. We compare pairwise similarities across all heads and layers using five distance metrics: Cosine similarity (COS), Pearson correlation (COR), Jensen–Shannon distance (JSD), Bhattacharyya coefficient (BCD), and Euclidean distance (ECD). Appendix D.1 presents qualitative visualizations for each metric to illustrate how they capture token-level relationships. Figure 6 reports their average accuracy

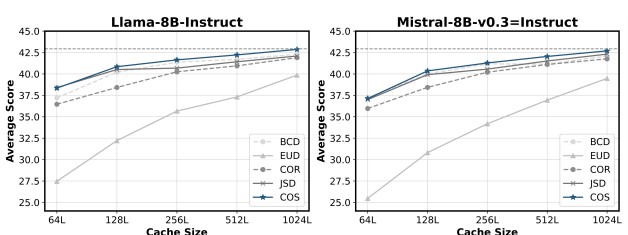

Figure 6: LongBench performance by cache budget for each distance matrix.

across all tasks under varying cache budgets. The results reveal that COS consistently achieves the highest performance in all settings, regardless of cache size. This indicates that COS not only provides a stable criterion for token retrieval in high-dimensional embedding spaces but also remains robust when computational resources are constrained. Taken together, these findings highlight the reliability of cosine similarity and motivate its use as the default metric throughout our experiments.

**Redundancy-Aware Allocation** *SCORE* adaptively allocates cache budgets across layers and heads based on quantified redundancy levels. As illustrated in Figure 5, components exhibiting greater information diversity tend to receive larger cache budgets, while highly redundant ones are assigned less. For instance, in Figure 11, lower layers tend to exhibit lower similarity to others, resulting in high $\mathcal{TD}$ in Figure 5(a), and are thus prioritized in layer-level budget assignment. Figure 5(b) further reveals that certain heads within these layers carry unique information, leading to increased head-level budget allocations. This analysis suggests that *SCORE* effectively leverages distance-based similarity metrics to identify structural redundancy and maximize informational diversity under constrained resources. Furthermore, Appendix D.3 validates the robustness and generality of *SCORE* through quantitative comparisons and visualizations.

## 4.4 EVALUATION ON LONG-CONTEXT TASKS

To further assess the generalizability of *SCORE*'s long-context reasoning, we conduct experiments on diverse benchmarks. Table 3 presents results on the Reasoning in a Haystack task under limited cache budgets and varying input lengths. *SCORE* consistently outperforms baselines across all lengths, indicating its ability to preserve precise reasoning. While prior methods often struggle as context grows, *SCORE* maintains strong performance, showing that its cache strategy—balancing importance and diversity—is effective for long-range dependency reasoning. Extended results on NeedleBench and InfiniteBench, provided in Appendix B and Appendix C, further support its robustness under diverse long-context conditions.

Table 3: Reasoning-in-a-Haystack results on Mistral-7B-Instruct with 128L KV cache. Scores are averaged over QA1–QA5 tasks at each context length.

| Method | 0k | 1k | 2k | 4k | 8k | 16k | 32k | Avg. |
|---|---|---|---|---|---|---|---|---|
| FullKV | 61.30 | 55.30 | 53.40 | 42.10 | 40.30 | 34.00 | 31.80 | 45.46 |
| SnapKV | 55.40 | 50.20 | 46.40 | 37.20 | 35.00 | 32.80 | 29.20 | 40.89 |
| PyramidKV | 57.20 | 50.80 | 47.60 | 36.20 | 36.20 | 31.40 | 28.20 | 41.09 |
| HeadKV | **58.60** | 53.80 | **52.20** | 38.20 | 37.60 | 31.80 | 30.40 | 43.23 |
| CAKE | 58.40 | 54.00 | 51.30 | **38.40** | 37.20 | 31.80 | 30.20 | 43.04 |
| SCORE | 58.40 | **54.20** | 51.80 | 38.30 | **37.80** | **32.10** | **30.60** | **43.31** |

## 4.5 EVALUATION ON MEMORY AND THROUGHPUT

To evaluate the practicality of *SCORE*, we compare it against prior methods in terms of time to first token (TTFT) and decoding latency. As shown in Figure 7, despite the need to compute a distance matrix, *SCORE* achieves comparable latency to existing approaches across all input lengths. This is notable given that *SCORE* estimates importance based on distance-based similarity between representations across layers and heads. Instead of computing full pairwise distances across all layers, *SCORE* adopts a cascading strategy that computes local distances

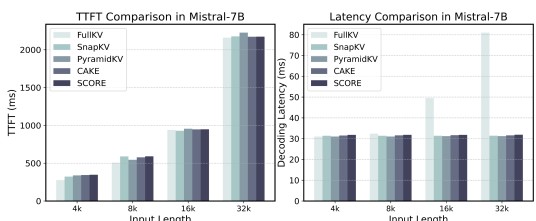

Figure 7: Comparison of TTFT (left) and decoding latency (right) across KV cache strategies in Mistral-7B-Instruct-v0.3.

only between adjacent layers, significantly reducing computation. Furthermore, the number of sample vectors used for similarity estimation is carefully controlled to further mitigate computational overhead. These results demonstrate that *SCORE* maintains responsiveness on par with existing methods, despite its more sophisticated scoring mechanism.

## 5 CONCLUSION

In this paper, we propose *SCORE*, a cache management framework that addresses structural redundancy and resource inefficiency in long-context processing. *SCORE* employs distance-based, multi-level similarity metrics to quantify representational redundancy across layers and attention heads. Using these estimates, it performs hierarchical budget reallocation and redundancy-aware token selection to preserve salient contextual information. To our knowledge, *SCORE* is the first method to directly measure redundancy for cache budgeting, enabling more effective modeling of inter-layer information flow than prior statistics-based approaches. Extensive experiments on long-context benchmarks demonstrate that *SCORE* consistently outperforms existing methods under tight cache constraints, particularly in tasks requiring complex reasoning and retrieval.

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

# APPENDIX

## A    MORE IMPLEMENTATION SETTINGS.

**Details of Hyper-parameters.** The algorithm of *SCORE* consists of three main stages: multi-level redundancy estimation, layer-wise budget allocation and cache management, and salient token selection. In the redundancy estimation stage (Section 3.2), we sample representations from each attention head and compute pairwise distances to extract multi-level redundancy. This process captures similarity patterns across layers, serving as the basis for quantifying cache necessity. In particular, we compute the $\mathcal{TD}$ between consecutive layers to assess the novelty of information introduced at each layer. To stabilize estimation, we apply EMA to $\mathcal{TD}$, searching $\gamma \in [0.1, 0.9]$ and setting it to 0.5. In the budget allocation and cache management stage (Section 3.3), we assign layer-wise KV cache budgets based on the estimated redundancy and perform eviction accordingly. We perform grid search over $\lambda_1, \lambda_2 \in [1.0, 2.0]$ with 0.2 step size

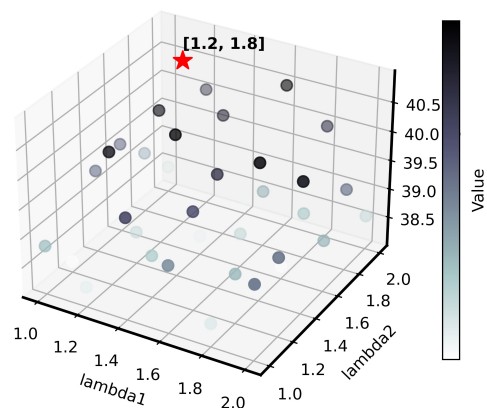

Figure 8: Performance comparison on LongBench for different $\lambda_1$ and $\lambda_2$ combinations.

for adaptive budget scaling. See Figure 8 for performance across different combinations. In the final stage (Section 3.4), we grid-search $\lambda_1, \lambda_2 \in [1.0, 2.0]$ (step 0.2) for adaptive scaling. This process is guided by a penalty term $\alpha$ that controls the preference for retaining high-redundancy tokens. The value of $\alpha$ is tuned within the range [0.001, 0.01], where larger values encourage more aggressive eviction of less salient tokens. We select $\alpha = 0.004$ based on this search.

**Analysis of sample selection.** To evaluate the impact of sample selection on assessment reliability, we fix a subset of 200 samples and compare three strategies: (1) top-k scoring, (2) uniform random sampling, and (3) mid-range random sampling, which excludes tokens from the initial and final context windows. As shown in Table 4, the mid-range strategy consistently yields more stable performance estimates. Excluding boundary regions mitigates evaluation artifacts caused by position-specific attention patterns. Mid-range sampling

Table 4: Performance comparison of sample selection strategies on LongBench using Llama3-8B under 128-cache. Reported values are average scores per task.

| Method | Single-Doc. | Multi-Doc. | Summ. | Few-shot | Synthetic | Code |
|---|---|---|---|---|---|---|
| Full | 32.97 | 33.65 | **22.39** | 66.92 | 38.99 | **55.17** |
| Random | **33.01** | 33.25 | 22.17 | 66.50 | 39.70 | 55.06 |
| Top-k | 31.10 | 29.11 | 20.74 | 65.20 | 32.74 | 53.39 |
| Middle | 32.72 | **33.67** | 22.30 | **67.72** | **39.75** | 54.89 |

reduces the influence of positional biases and better captures model behavior in regions where memory and generalization demands are more representative of typical usage.

**Ablation on Sample Size.** To keep computation cost stable regardless of sequence length, we fix the number of sampled tokens $K$ when estimating head-wise similarity. Nevertheless, in extremely long-context scenarios, model performance could in principle depend on the choice of $K$. Table 5 therefore presents an ablation on Llama3-8B-Instruct with LongBench V2 (input sequences extended up to $\sim$2M tokens) under a cache size of 128, comparing SCORE with $K = 200$ and $K = 400$ against baseline methods. The results show that the performance gap between different $K$ values remains minimal, confirming that even a relatively small $K$ is sufficient and that SCORE is robust to the sampling.

Table 5: Ablation study on Llama3-8B-Instruct with LongBench V2. We analyze the effect of different sample sizes ($K = 200$ and $K = 400$) for our method (SCORE) under cache size 128, compared against CAKE and HeadKV.

| Method | Easy | Hard | Short | Medium | Long | Avg. |
|---|---|---|---|---|---|---|
| Full KV | 31.25 | 25.08 | 34.44 | 24.19 | 22.22 | 27.44 |
| Llama3-8B-Instruct, Cache size=128 | | | | | | |
| CAKE | 29.27 | 21.13 | **32.98** | 21.22 | 20.18 | 24.96 |
| HeadKV | 29.53 | 22.74 | 32.25 | 22.97 | 20.10 | 25.52 |
| SCORE ($K = 200$) | **29.94** | 23.11 | 32.78 | 23.73 | 20.33 | 25.98 |
| SCORE ($K = 400$) | 29.85 | **23.40** | 32.71 | **23.80** | **20.61** | **26.07** |

## B    EXPERIMENTS ON NEEDLEBENCH DATASET

To assess the retrieval capabilities of our proposed method, *SCORE*, we conduct comprehensive evaluations on the Needle-in-a-Haystack benchmark. This benchmark is specifically designed to test a model's ability to accurately identify and extract salient information (needle) from extensive input sequences (haystack). We evaluate on both LLaMA-3-8B-Instruct and Mistral-7B-Instruct-v0.3, setting the maximum context lengths to 8K and 32K tokens, respectively, as summarized in Figure 9,10. At a cache size of 128, *SCORE* demonstrates strong retrieval fidelity in short-context settings while exhibiting minimal degradation in performance under long-context conditions. Notably, *SCORE* achieves an accuracy of 96.7 on the Mistral-7B-Instruct-v0.3 model, outperforming the previous state-of-the-art method, HeadKV, which attains 95.5—representing a relative improvement of 1.2 points. Similarly, on Llama3-8B-Instruct, *SCORE* consistently matches or exceeds the performance of existing methods across varying context lengths. These results demonstrate that *SCORE* is highly capable of retrieving and processing salient information even under ultra-long context conditions. Importantly, *SCORE* maintains robust retrieval performance even when the KV cache size is significantly reduced (*e.g.*, 128), with only negligible accuracy degradation compared to FullKV. This highlights the effectiveness of *SCORE* in balancing memory efficiency and performance, offering a promising solution for memory-constrained long-context language modeling.

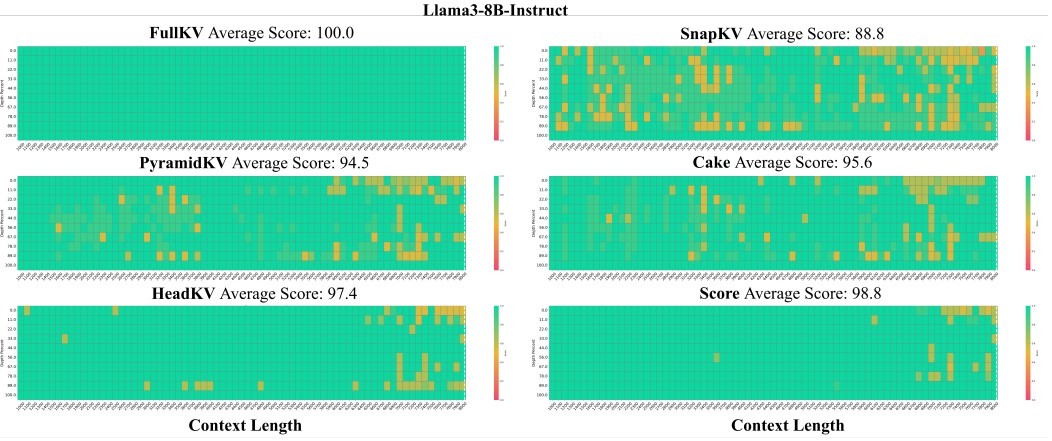

Figure 9: Needle-in-a-Haystack test results on Llama-3-8B-Instruct with KV cache = 128. Our proposed SCORE method significantly outperform all strong baselines.

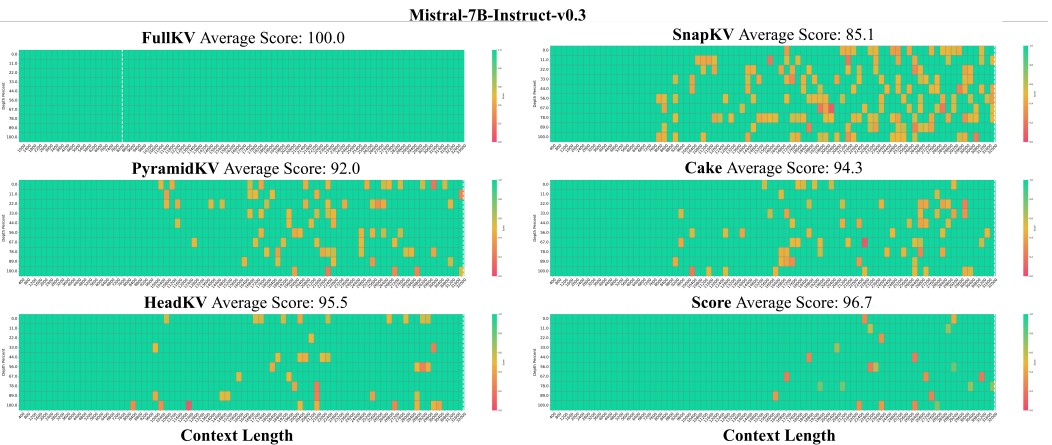

Figure 10: Needle-in-a-Haystack test results on Mistral-7B-Instruct-v0.3 with KV cache = 128. Our proposed SCORE method significantly outperform all strong baselines.

# C  EXPERIMENTS ON INFINITEBENCH DATASET

In this section, we evaluate the effectiveness of the proposed method, *SCORE*, on InfiniteBench—a challenging benchmark specifically designed to assess the long-context processing capabilities of large language models. A detailed comparison of results across models and cache budgets is provided in Table 6,7. Compared to prior benchmarks such as LongBench, InfiniteBench introduces substantially greater difficulty by incorporating ultra-long input sequences, with an average context length of 145K tokens and a maximum of up to 214K tokens. The benchmark spans five diverse domains—Retrieval, Code, Math, Novels, and Dialogue—providing a comprehensive testbed for evaluating a model's ability to understand, reason over, and extract salient information from extremely long contexts. We conduct experiments using both Llama3-8B-Instruct and Mistral-7B-Instruct-v0.3, evaluating performance across 10 datasets from InfiniteBench under two KV cache budgets: 128L and 1024L. At a cache size of 128L, *SCORE* achieves the highest average accuracy of 17.98 on Mistral-7B-Instruct-v0.3, outperforming the previous best-performing method, CAKE, which attains 17.57—a margin of 0.41. When the cache size is increased to 1024L, *SCORE* further improves to 19.54, again surpassing CAKE (18.99), with a larger margin of 0.55. These results demonstrate the effectiveness of *SCORE* in compressing and retaining salient information within memory-constrained settings, even under extreme sequence lengths. Importantly, *SCORE* maintains consistent performance improvements across both low and high cache budgets, underscoring its robustness and scalability. This highlights *SCORE* as a compelling solution for long-context language modeling, effectively balancing memory efficiency with task performance across a wide range of domains and input lengths.

Table 6: Performance comparison over 10 datasets of InfiniteBench on Llama3-8B-Instruct and Mistral-7B-Instruct-v0.3. Results are measured with 128L cache. The best result is highlighted in **bold**, the second best in underline.

| | Method | Retrieval | | Code | | Math | Novels | | | | Dialogue |
| | | Ret.PassKey | Ret.Number | Code.Debug | Code.Run | Math.Find | En.Sum | En.QA | En.choice | Chn.QA | En.Dia |
|---|---|---|---|---|---|---|---|---|---|---|---|
| **Llama3-8B-Instruct** | StreamingLLM | 4.98 | 4.46 | 39.00 | 0.50 | 19.64 | 19.49 | 0.01 | 41.06 | 1.94 | 0.00 |
| | H2O | 5.30 | 5.57 | 40.19 | 0.50 | 20.32 | 20.01 | 0.01 | 41.16 | 2.21 | 0.00 |
| | HeadKV | 6.14 | 6.14 | 43.54 | 1.50 | 26.51 | 23.37 | 0.09 | 47.43 | 3.67 | 0.00 |
| | SnapKV | 6.34 | 6.32 | 43.32 | 1.50 | 25.83 | 23.45 | 0.10 | 48.25 | 3.41 | 0.00 |
| | PyramidKV | 6.48 | 6.21 | 43.44 | 1.50 | 26.24 | 23.32 | **0.16** | 48.03 | 3.16 | **0.50** |
| | CAKE | 6.22 | **6.51** | 43.48 | 1.50 | 26.74 | **25.11** | 0.05 | 48.16 | 3.75 | **0.50** |
| | SCORE(ours) | **6.53** | 5.93 | **43.61** | **2.00** | **26.86** | 23.78 | 0.03 | **48.24** | **3.98** | **0.50** |
| **Mistral-7B-Instruction-v0.3** | StreamingLLM | 24.10 | 5.56 | 28.17 | 0.00 | 21.56 | 20.03 | 0.10 | 40.56 | 8.01 | 0.00 |
| | H2O | 25.02 | 5.63 | 29.28 | 0.25 | 22.74 | 20.12 | 0.10 | 40.98 | 7.89 | 0.00 |
| | HeadKV | 26.75 | 6.27 | 32.74 | 0.25 | **27.43** | 22.74 | 0.31 | 49.10 | **10.32** | 0.50 |
| | SnapKV | 25.95 | 6.95 | 31.47 | **0.50** | 22.57 | **23.14** | 0.32 | 48.76 | 9.69 | 0.00 |
| | PyramidKV | 26.78 | 6.69 | **33.25** | 0.25 | 23.43 | 21.92 | 0.28 | 49.53 | 9.01 | 0.00 |
| | CAKE | 26.68 | 6.64 | 32.49 | **0.50** | 28.14 | 21.45 | 0.33 | 49.56 | 9.93 | 0.00 |
| | SCORE(ours) | **26.89** | **7.01** | 32.89 | **0.50** | 28.57 | 22.29 | **0.39** | **50.16** | 10.16 | **1.00** |

Table 7: Performance comparison over 10 datasets of InfiniteBench on Llama3-8B-Instruct and Mistral-7B-Instruct-v0.3. Results are measured with 1024L cache. The best result is highlighted in **bold**, the second best in underline.

| | Method | Retrieval | | Code | | Math | Novels | | | | Dialogue |
| | | Ret.PassKey | Ret.Number | Code.Debug | Code.Run | Math.Find | En.Sum | En.QA | En.choice | Chn.QA | En.Dia |
|---|---|---|---|---|---|---|---|---|---|---|---|
| **Llama3-8B-Instruct** | StreamingLLM | 5.89 | 5.78 | 40.78 | 1.50 | 25.21 | 20.79 | 0.05 | 41.87 | 2.78 | 0.50 |
| | H2O | 5.48 | 6.14 | 42.23 | 1.00 | 25.49 | 21.31 | 0.05 | 42.65 | 3.12 | 0.50 |
| | HeadKV | 6.73 | 6.61 | 44.53 | 2.00 | 26.98 | 25.10 | 0.04 | 49.87 | 3.77 | 0.50 |
| | SnapKV | 6.42 | 6.58 | 44.91 | 2.50 | 27.13 | 24.64 | 0.05 | 48.42 | 3.34 | 0.50 |
| | PyramidKV | 6.59 | 6.44 | 44.87 | 2.25 | 27.17 | 25.00 | 0.15 | 49.47 | 4.10 | **1.00** |
| | CAKE | 6.71 | 6.76 | 44.79 | 2.50 | **27.43** | 24.28 | 0.10 | 50.12 | 3.68 | 0.50 |
| | SCORE(ours) | **6.78** | **6.78** | **45.16** | **3.00** | **27.43** | **25.12** | **0.25** | **50.66** | **4.14** | **1.00** |
| **Mistral-7B-Instruction-v0.3** | StreamingLLM | 26.00 | 8.51 | 30.28 | 0.50 | 25.67 | 21.01 | 0.05 | 47.13 | 8.78 | 0.00 |
| | H2O | 26.21 | 8.98 | 29.78 | 0.50 | 26.97 | 20.31 | 0.09 | 48.78 | 9.24 | 0.00 |
| | HeadKV | 27.00 | 10.12 | 35.28 | 1.00 | 28.71 | 21.65 | **0.45** | 50.51 | 10.48 | 0.00 |
| | SnapKV | 26.98 | 9.37 | 33.25 | 1.00 | 30.29 | 21.68 | 0.19 | 49.79 | 10.59 | 1.00 |
| | PyramidKV | 26.45 | 9.49 | 34.26 | 1.00 | 26.86 | 22.24 | 0.33 | 49.65 | 10.54 | 1.00 |
| | CAKE | 27.02 | 12.15 | 36.80 | 1.00 | 29.71 | 22.27 | 0.33 | 49.23 | 10.34 | 1.00 |
| | SCORE(ours) | **27.12** | **12.20** | **39.09** | **1.50** | **30.71** | **22.68** | 0.28 | **50.66** | **10.65** | **1.50** |

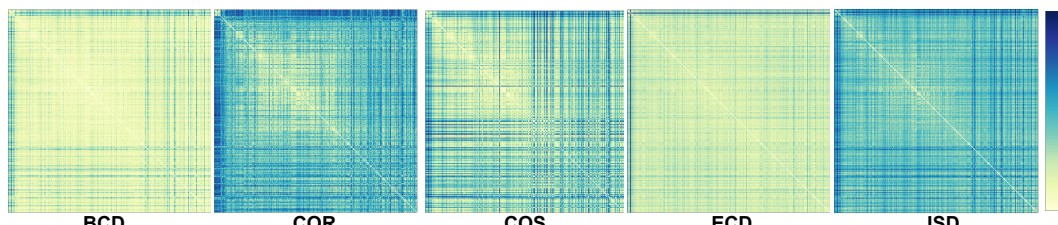

Figure 11: Head-to-head similarity matrices across various metrics for NarrativeQA. Each value represents the average distance between 200 sampled attention vectors per head.

# D    ADDITIONAL DETAILED ANALYSIS AND VISUALIZATION

## D.1    EFFECT OF DISTANCE METRIC CHOICE

Figure 11 visualizes pairwise similarities across all heads and layers using various distance metrics: Cosine similarity (COS), Pearson correlation (COR), Jensen–Shannon distance (JSD), Bhattacharyya coefficient (BCD), and Euclidean distance (ECD). Bright regions indicate high similarity, with all metrics revealing consistent alignment across specific layers and heads. These metrics span a diverse range of similarity formulations, from distribution-based measures (JSD, BCD) to correlation- or geometry-based ones (COR, EUD), providing a comprehensive comparison. Despite differing statistical bases, these metrics capture similar redundancy signals, suggesting pronounced structural redundancy in model representations. This highlights the potential of redundancy-aware selection to improve efficiency.

## D.2    VISUALIZATION OF DISTANCE MATRIX

To further analyze the behavior of the cosine distance metric in our retrieval framework, we visualize the pairwise distance matrices computed using cosine similarity across different datasets in LongBench, as shown in Figure 12. Each matrix represents the inter-token similarity structure within heads, with darker regions indicating lower similarity (*i.e.*, higher cosine distance). Across datasets, we observe consistent patterns of redundancy, where certain groups of heads exhibit strong mutual similarity. While the specific patterns vary depending on the dataset domain and structure, the presence of high-similarity clusters is a common characteristic. These clusters often correspond to repeated representations. Such redundancy can degrade retrieval efficiency and content diversity if not properly managed. *SCORE* addresses this by leveraging cosine distance not only to capture salient content but also to suppress over-represented or semantically repetitive tokens. This behavior is especially beneficial in budget-constrained settings, where the selection of maximally informative yet diverse tokens is critical.

## D.3    VISUALIZATION OF BUDGET ALLOCATION

To better understand how retrieval budgets are distributed across the model's architecture, we visualize intra-layer similarity patterns across different datasets in LongBench, as shown in Figure 13. Each heatmap captures the pairwise similarity within heads at each layer, providing insight into redundancy and representational diversity. We observe a consistent trend across datasets: the lower layers generally exhibit higher diversity, as indicated by lower intra-head similarity. These layers tend to capture localized, fine-grained features, making their token representations less redundant. Consequently, they receive a larger share of the retrieval budget, allowing more tokens to be selected from them. In contrast, middle layers often show pronounced redundancy, with many heads producing highly similar token embeddings. The final column in Figure 13 further breaks down the intra-layer budget allocation at the head level. Within a given layer, heads exhibiting high distinctiveness are allocated more budget. This fine-grained allocation strategy ensures that the most informative and non-redundant heads are prioritized, aligning with the principles described in Section 3.3.

Despite variations in domain and task, several datasets exhibit notably similar similarity profiles across layers and heads, suggesting that token representation patterns are influenced not only by data but also by the model's inherent architecture. This structural consistency points to a promising direction for developing more generalized, architecture-aware retrieval strategies.

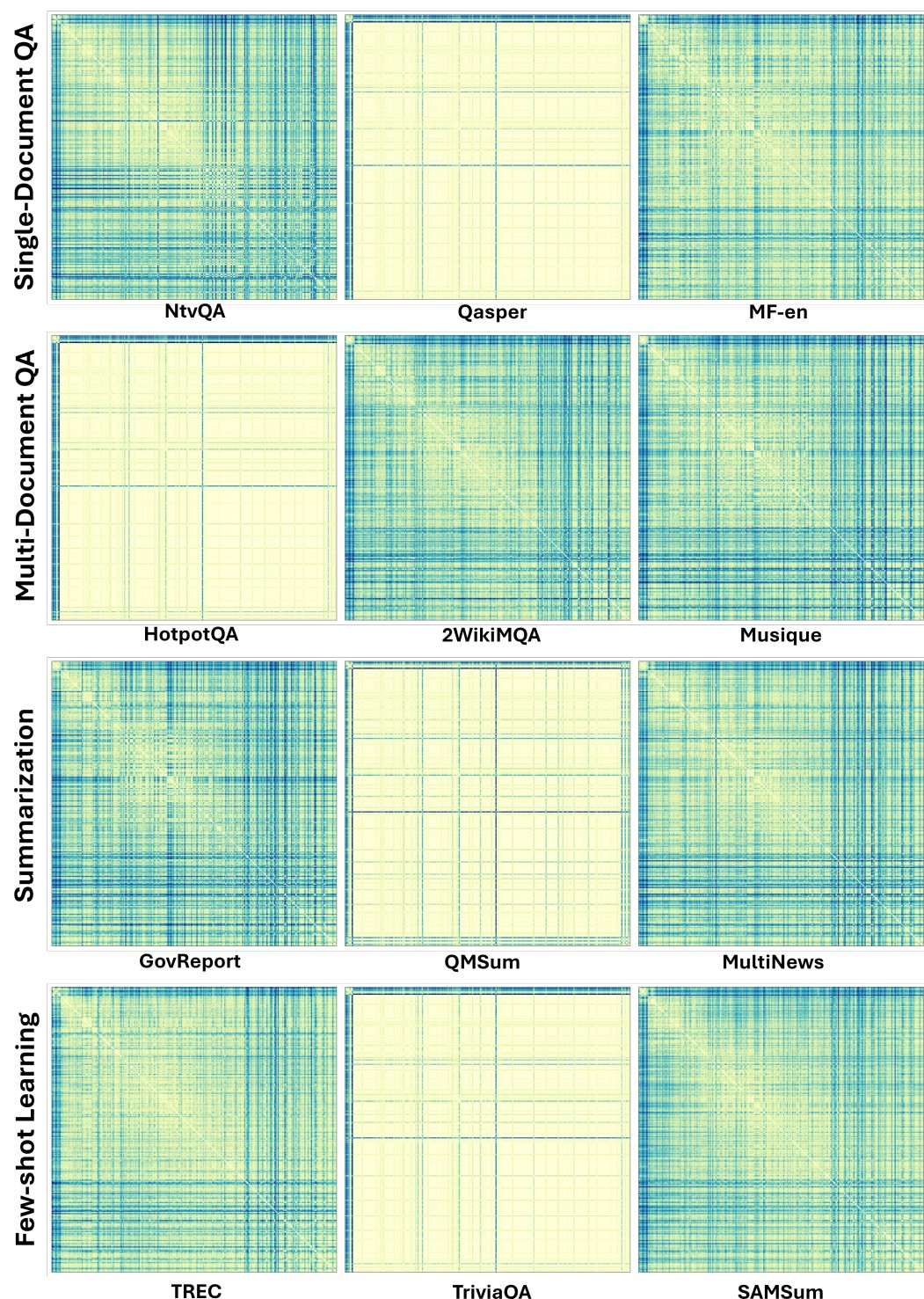

Figure 12: Cosine distance matrices for various datasets in LongBench.

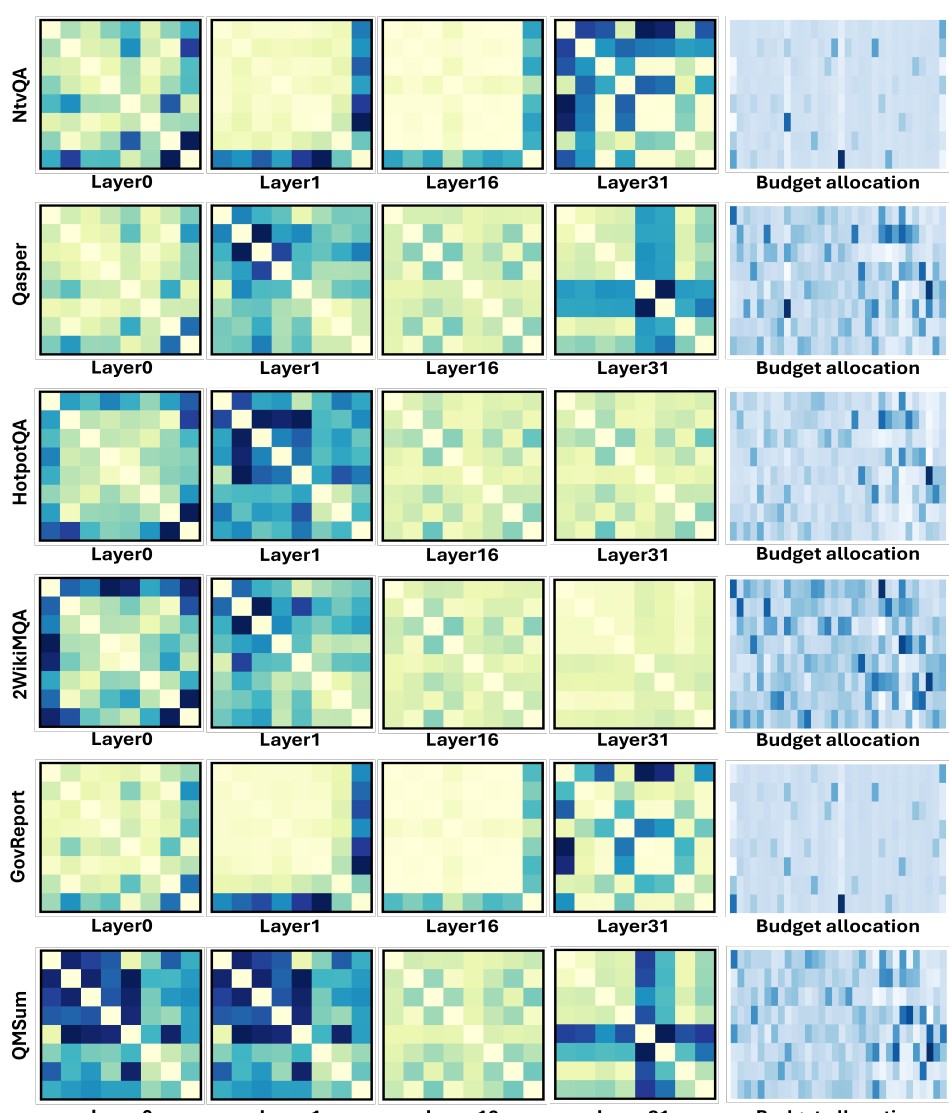

Figure 13: Cosine distance matrices for various datasets in LongBench.

# E    EXPERIMENTS ON LONGBENCH DATASET

We provide a full breakdown of the LongBench evaluation results for LlaMA-2-7B, LlaMA-2-13B, LlaMA-3-8B-Instruct, and Mistral-7B-v0.3-Instruct. The results are presented in ascending order of cache size: 64L (Table 8), 256L (Table 9), 512L (Table 10), and 1024L (Table 11).

**Results on Llama3-8B-Instruct.** *SCORE* consistently outperforms prior works across all cache sizes on the LongBench benchmark. The advantage is especially prominent in the low-cache regime, notably at 64L (Table 8), where *SCORE* demonstrates a significantly higher score compared to other methods. A key observation is that existing works often select redundant key-value pairs under tight cache budgets, leading to poor coverage and lower performance. In contrast, SCORE maintains accuracy even at 64L through diversity-aware selection, yielding more informative memory allocation.

**Results on Mistral-7B-Instruct-v0.3.** Similar to the observations with Llama3-8B-Instruct, our method (SCORE) shows strong and stable performance across cache sizes when evaluated on Mistral-7B-Instruct-v0.3. Notably, even under the extreme low-cache setting of 64L, *SCORE* preserves 85.9% of its performance relative to the full-cache setting, demonstrating robustness.

Table 8: Performance comparison over 16 datasets from LongBench. Results are measured with a cache size of 64. The best score is marked in **bold**, and the second best is marked with underline.

| | Method | Single-Document QA | | | Multi-Document QA | | | Summarization | | | Few-shot Learning | | | Synthetic | | Code | |
|---|---|---|---|---|---|---|---|---|---|---|---|---|---|---|---|---|---|
| | | NtvQA | Qasper | MF-en | HotpotQA | 2WikiMQA | Musique | GovReport | QMSum | MultiNews | TREC | TriviaQA | SAMSum | PCount | PR-en | Lcc | RB-P |
| Llama2-7B | StreamingLLM | 10.70 | 15.36 | 27.88 | 24.20 | 20.35 | 5.89 | 14.85 | 16.23 | 18.40 | 55.00 | 78.85 | 34.75 | 2.50 | 1.50 | 50.25 | 44.65 |
| | H2O | 11.33 | 17.26 | 29.21 | 24.49 | 21.46 | 7.84 | 16.49 | 18.00 | 20.00 | 62.00 | 80.30 | 36.29 | 5.00 | 2.50 | 51.46 | 46.19 |
| | HeadKV | 14.41 | 18.80 | 27.95 | 26.94 | 23.85 | 7.93 | 17.84 | 19.56 | 20.65 | 63.00 | 81.54 | 37.47 | 5.50 | 4.00 | 53.09 | 47.89 |
| | SnapKV | 13.67 | 18.05 | 26.80 | 28.00 | 21.38 | 7.49 | 16.78 | 19.59 | 19.85 | 57.50 | 81.40 | 36.98 | 5.50 | 3.50 | 51.60 | 45.96 |
| | PyramidKV | 14.75 | 18.25 | 28.57 | 29.27 | 23.12 | 7.50 | 17.42 | 19.56 | 19.87 | 60.00 | 82.14 | 36.26 | 5.50 | 3.50 | 52.13 | 46.23 |
| | CAKE | 15.49 | 18.88 | 31.97 | 29.33 | 24.65 | 9.35 | 18.62 | 19.88 | 22.21 | 63.00 | 82.82 | 38.62 | 6.00 | 6.00 | 54.46 | 48.53 |
| | SCORE(ours) | 15.48 | 18.75 | 32.12 | 29.01 | 23.91 | 10.01 | 18.43 | 20.02 | 20.29 | 63.00 | 83.47 | 36.67 | 6.00 | 5.50 | 53.77 | 47.11 |
| Llama2-13B | StreamingLLM | 10.45 | 7.56 | 21.95 | 9.03 | 7.13 | 1.61 | 17.45 | 16.75 | 17.73 | 60.00 | 82.10 | 37.31 | 0.50 | 12.50 | 32.45 | 35.89 |
| | H2O | 11.84 | 9.31 | 23.36 | 9.12 | 9.00 | 3.26 | 18.88 | 19.39 | 19.93 | 59.50 | 85.94 | 37.86 | 1.50 | 14.72 | 37.67 | 36.07 |
| | HeadKV | 11.99 | 12.14 | 23.53 | 10.65 | 9.03 | 3.70 | 19.28 | 19.86 | 19.85 | 64.00 | 84.74 | 38.48 | 2.50 | 15.75 | 35.92 | 36.22 |
| | SnapKV | 11.51 | 10.87 | 22.92 | 10.24 | 8.84 | 3.20 | 18.60 | 19.25 | 19.19 | 50.50 | 86.84 | 36.28 | 2.00 | 16.75 | 34.97 | 33.60 |
| | PyramidKV | 12.55 | 11.44 | 24.53 | 10.84 | 10.04 | 4.01 | 18.92 | 19.83 | 20.15 | 57.00 | 86.31 | 38.18 | 2.00 | 16.25 | 37.19 | 35.49 |
| | CAKE | 12.69 | 10.11 | 22.41 | 10.21 | 9.70 | 2.81 | 20.03 | 20.20 | 21.63 | 67.50 | 87.12 | 39.29 | 2.00 | 15.75 | 42.76 | 41.25 |
| | SCORE(ours) | 12.96 | 12.44 | 24.20 | 10.56 | 9.73 | 3.70 | 20.37 | 20.27 | 22.18 | 68.00 | 87.50 | 40.13 | 2.00 | 16.75 | 43.22 | 40.61 |
| Llama3-8B-Instruct | StreamingLLM | 20.26 | 16.29 | 30.02 | 34.81 | 25.73 | 13.70 | 15.26 | 19.30 | 12.94 | 60.50 | 80.77 | 30.62 | 4.63 | 64.00 | 48.87 | 43.84 |
| | H2O | 20.76 | 17.78 | 32.43 | 38.28 | 27.08 | 16.01 | 17.72 | 20.61 | 16.36 | 61.50 | 88.26 | 35.95 | 4.88 | 68.00 | 50.56 | 48.41 |
| | HeadKV | 23.67 | 17.46 | 32.72 | 39.81 | 27.50 | 17.39 | 19.14 | 22.12 | 20.04 | 65.00 | 90.20 | 37.27 | 4.77 | 69.00 | 54.68 | 51.52 |
| | SnapKV | 21.56 | 17.98 | 32.35 | 38.45 | 26.40 | 17.01 | 18.38 | 21.99 | 18.86 | 51.50 | 89.32 | 36.11 | 5.00 | 69.50 | 53.13 | 49.81 |
| | PyramidKV | 21.76 | 18.56 | 33.01 | 39.45 | 28.47 | 17.42 | 18.58 | 21.77 | 18.58 | 58.50 | 88.14 | 37.26 | 5.00 | 69.50 | 52.05 | 47.27 |
| | CAKE | 22.06 | 18.17 | 32.98 | 40.53 | 31.56 | 17.98 | 19.94 | 22.30 | 21.27 | 70.50 | 90.40 | 38.85 | 6.00 | 69.50 | 54.62 | 49.75 |
| | SCORE(ours) | 22.68 | 18.77 | 33.26 | 41.06 | 31.58 | 17.84 | 20.41 | 22.20 | 21.58 | 72.00 | 91.00 | 38.78 | 5.50 | 70.00 | 56.08 | 50.69 |
| Mistral-7B-Instruction-v0.3 | StreamingLLM | 19.54 | 25.07 | 37.66 | 40.13 | 21.43 | 12.69 | 17.47 | 16.93 | 15.99 | 51.00 | 84.00 | 37.20 | 4.00 | 70.00 | 41.44 | 39.34 |
| | H2O | 20.10 | 25.40 | 37.92 | 40.48 | 21.78 | 13.00 | 17.73 | 17.51 | 16.37 | 57.00 | 84.50 | 37.62 | 4.00 | 75.00 | 41.83 | 39.96 |
| | HeadKV | 23.18 | 26.31 | 38.77 | 40.94 | 21.20 | 13.38 | 18.51 | 17.68 | 16.74 | 62.50 | 85.00 | 38.47 | 4.50 | 69.50 | 42.35 | 41.38 |
| | SnapKV | 22.21 | 26.53 | 37.83 | 39.96 | 20.10 | 13.00 | 17.20 | 17.33 | 16.00 | 50.00 | 85.00 | 36.54 | 5.00 | 76.50 | 41.22 | 38.60 |
| | PyramidKV | 21.89 | 26.53 | 37.90 | 40.52 | 22.17 | 13.41 | 17.78 | 17.69 | 19.66 | 54.50 | 84.00 | 36.56 | 5.50 | 77.50 | 40.16 | 37.42 |
| | CAKE | 23.89 | 26.89 | 39.12 | 42.08 | 25.26 | 13.97 | 19.38 | 18.03 | 18.25 | 64.50 | 85.00 | 40.22 | 5.00 | 78.00 | 45.02 | 42.94 |
| | SCORE(ours) | 23.77 | 27.86 | 38.81 | 42.12 | 25.28 | 13.83 | 19.07 | 18.98 | 18.38 | 67.00 | 85.50 | 40.54 | 5.00 | 78.50 | 45.22 | 44.10 |

Table 9: Performance comparison over 16 datasets from LongBench. Results are measured with a cache size of 256. The best score is marked in **bold**, and the second best is marked with underline.

| | Method | Single-Document QA | | | Multi-Document QA | | | Summarization | | | Few-shot Learning | | | Synthetic | | Code | |
|---|---|---|---|---|---|---|---|---|---|---|---|---|---|---|---|---|---|
| | | NtvQA | Qasper | MF-en | HotpotQA | 2WikiMQA | Musique | GovReport | QMSum | MultiNews | TREC | TriviaQA | SAMSum | PCount | PR-en | Lcc | RB-P |
| Llama2-7B | StreamingLLM | 15.09 | 18.87 | 29.73 | 29.60 | 23.70 | 8.72 | 18.69 | 19.32 | 20.98 | 62.50 | 82.69 | 34.00 | 3.00 | 6.00 | 55.04 | 48.80 |
| | H2O | 15.63 | 19.41 | 30.27 | 30.14 | 24.24 | 9.26 | 19.23 | 19.86 | 21.52 | 63.00 | 83.23 | 34.54 | 3.50 | 6.50 | 55.58 | 49.34 |
| | HeadKV | 15.66 | 20.29 | 30.95 | 30.52 | 24.50 | 9.87 | 19.78 | 20.33 | 21.93 | 64.00 | 83.19 | 35.18 | 5.00 | 5.50 | 56.01 | 50.13 |
| | SnapKV | 16.10 | 20.16 | 30.59 | 30.44 | 24.89 | 9.78 | 20.15 | 20.37 | 21.78 | 63.50 | 83.90 | 34.78 | 4.50 | 6.00 | 56.97 | 49.98 |
| | PyramidKV | 16.32 | 18.99 | 30.47 | 30.65 | 24.53 | 9.32 | 18.95 | 20.07 | 22.05 | 63.00 | 83.80 | 34.85 | 5.00 | 10.00 | 56.97 | 49.12 |
| | CAKE | 15.91 | 19.69 | 30.55 | 30.42 | 24.52 | 9.54 | 19.51 | 20.14 | 21.80 | 63.00 | 83.51 | 34.82 | 4.50 | 7.00 | 55.86 | 49.62 |
| | SCORE(ours) | 17.24 | 21.35 | 35.52 | 30.69 | 25.43 | 9.76 | 20.52 | 20.46 | 23.83 | 64.00 | 84.30 | 34.98 | 5.00 | 9.50 | 57.45 | 51.49 |
| Llama2-13B | StreamingLLM | 12.45 | 13.66 | 25.71 | 11.03 | 9.27 | 3.81 | 20.97 | 19.47 | 22.54 | 68.00 | 85.42 | 44.77 | 4.50 | 12.00 | 42.88 | 41.58 |
| | H2O | 12.99 | 14.20 | 26.25 | 11.57 | 9.81 | 4.35 | 21.51 | 20.01 | 23.08 | 68.00 | 85.96 | 45.31 | 4.00 | 14.75 | 43.42 | 42.12 |
| | HeadKV | 13.77 | 15.17 | 27.50 | 11.94 | 10.48 | 4.01 | 21.78 | 20.54 | 23.57 | 69.00 | 86.83 | 46.38 | 5.50 | 14.25 | 43.95 | 42.69 |
| | SnapKV | 12.56 | 12.82 | 26.13 | 12.49 | 10.38 | 4.85 | 22.61 | 20.44 | 23.56 | 69.00 | 86.15 | 45.82 | 3.05 | 14.75 | 44.33 | 43.36 |
| | PyramidKV | 13.85 | 15.82 | 26.33 | 11.49 | 9.76 | 5.38 | 21.35 | 20.24 | 23.30 | 69.00 | 86.09 | 44.94 | 5.57 | 16.75 | 43.18 | 41.51 |
| | CAKE | 13.27 | 14.48 | 26.53 | 11.85 | 10.09 | 4.63 | 21.79 | 20.29 | 23.36 | 68.50 | 86.24 | 45.59 | 4.15 | 15.25 | 43.70 | 42.40 |
| | SCORE(ours) | 13.68 | 16.58 | 28.97 | 12.70 | 12.34 | 4.77 | 22.38 | 20.46 | 24.67 | 69.00 | 87.56 | 47.16 | 5.50 | 16.25 | 44.25 | 43.38 |
| Llama3-8B-Instruct | StreamingLLM | 22.87 | 31.41 | 40.45 | 43.13 | 31.64 | 19.79 | 20.69 | 21.66 | 21.88 | 70.50 | 89.70 | 39.24 | 4.50 | 68.48 | 56.46 | 52.83 |
| | H2O | 23.41 | 31.95 | 40.99 | 43.67 | 32.18 | 20.33 | 21.23 | 22.20 | 22.42 | 71.00 | 90.24 | 39.78 | 5.00 | 69.02 | 57.00 | 53.37 |
| | HeadKV | 24.08 | 34.01 | 42.64 | 44.52 | 33.54 | 21.31 | 21.79 | 22.33 | 23.09 | 72.00 | 90.57 | 40.62 | 5.75 | 69.50 | 57.68 | 55.29 |
| | SnapKV | 23.39 | 33.50 | 40.73 | 43.80 | 32.84 | 20.10 | 21.69 | 22.57 | 22.79 | 71.50 | 90.86 | 39.85 | 5.51 | 69.50 | 58.14 | 53.82 |
| | PyramidKV | 23.95 | 29.53 | 40.81 | 43.90 | 31.37 | 20.77 | 21.42 | 22.89 | 22.59 | 71.50 | 90.48 | 40.08 | 5.91 | 69.25 | 56.37 | 52.20 |
| | CAKE | 23.71 | 32.25 | 41.29 | 43.97 | 32.48 | 20.63 | 21.53 | 22.50 | 22.72 | 71.50 | 90.54 | 40.08 | 5.36 | 69.32 | 57.30 | 53.67 |
| | SCORE(ours) | 25.09 | 35.68 | 42.93 | 45.61 | 35.63 | 21.37 | 22.80 | 22.69 | 24.25 | 72.50 | 90.56 | 41.98 | 5.78 | 69.25 | 57.23 | 53.01 |
| Mistral-7B-Instruction-v0.3 | StreamingLLM | 26.00 | 30.63 | 49.01 | 43.47 | 31.20 | 22.43 | 20.29 | 18.32 | 19.01 | 67.00 | 84.42 | 40.70 | 4.50 | 79.39 | 45.22 | 45.79 |
| | H2O | 26.54 | 31.17 | 49.55 | 44.01 | 31.74 | 22.97 | 20.83 | 18.86 | 19.55 | 68.00 | 84.96 | 41.24 | 5.00 | 79.93 | 45.76 | 46.33 |
| | HeadKV | 27.66 | 32.30 | 49.98 | 44.79 | 32.35 | 23.32 | 20.99 | 20.16 | 20.03 | 69.50 | 85.30 | 42.35 | 6.00 | 79.50 | 46.96 | 48.83 |
| | SnapKV | 26.55 | 31.24 | 49.58 | 43.57 | 31.80 | 23.14 | 21.44 | 18.93 | 20.03 | 68.00 | 85.01 | 41.36 | 5.00 | 81.00 | 46.45 | 46.85 |
| | PyramidKV | 26.61 | 31.18 | 50.28 | 44.87 | 32.28 | 23.65 | 21.27 | 18.69 | 19.80 | 67.50 | 85.78 | 41.22 | 5.00 | 80.50 | 45.07 | 44.50 |
| | CAKE | 26.34 | 31.99 | 49.39 | 44.02 | 31.80 | 23.28 | 21.28 | 18.86 | 19.90 | 68.00 | 85.17 | 41.54 | 5.00 | 80.77 | 46.21 | 46.26 |
| | SCORE(ours) | 27.05 | 34.74 | 49.52 | 44.81 | 32.53 | 24.26 | 22.32 | 20.15 | 21.06 | 70.50 | 85.91 | 43.23 | 5.50 | 82.00 | 48.32 | 48.63 |

Table 10: Performance comparison over 16 datasets from LongBench. Results are measured with a cache size of 512. The best score is marked in **bold**, and the second best is marked with underline.

| Model | Method | Single-Document QA | | | Multi-Document QA | | | Summarization | | | Few-shot Learning | | | Synthetic | | Code | |
|---|---|---|---|---|---|---|---|---|---|---|---|---|---|---|---|---|---|
| | | NtvQA | Qasper | MF-en | HotpotQA | 2WikiMQA | Musique | GovReport | QMSum | MultiNews | TREC | TriviaQA | SAMSum | PCount | PR-en | Lcc | RB-P |
| Llama2-7B | StreamingLLM | 15.94 | 19.52 | 33.89 | 30.44 | 24.12 | 9.20 | 20.32 | 19.56 | 22.40 | 63.00 | 82.56 | 34.29 | 4.00 | 6.00 | 56.85 | 50.41 |
| | H2O | 16.48 | 20.06 | 34.43 | 30.98 | 24.66 | 9.74 | 20.86 | 20.10 | 22.94 | 63.00 | 83.10 | 34.83 | 4.50 | 6.50 | 57.39 | 50.95 |
| | HeadKV | 17.02 | 21.38 | 35.27 | 31.48 | 25.08 | 9.50 | 21.59 | 20.76 | 22.59 | 64.00 | 83.02 | 34.77 | 6.00 | 5.50 | 57.98 | 51.65 |
| | SnapKV | 17.02 | 19.96 | 34.18 | 31.37 | 24.95 | 10.51 | 21.74 | 20.22 | 23.79 | 64.00 | 83.57 | 35.12 | 5.00 | 7.00 | 58.54 | 50.98 |
| | PyramidKV | 16.60 | 20.03 | 35.03 | 31.30 | 25.16 | 10.40 | 20.46 | 20.52 | 23.63 | 64.00 | 83.90 | 35.80 | 5.00 | 8.00 | 56.85 | 51.41 |
| | CAKE | 16.76 | 20.34 | 34.71 | 31.26 | 24.94 | 10.02 | 21.14 | 20.38 | 23.22 | 63.88 | 83.38 | 35.11 | 4.50 | 7.05 | 57.67 | 51.23 |
| | SCORE(ours) | 18.29 | 21.57 | 35.95 | 30.78 | 24.88 | 10.84 | 22.08 | 20.74 | 24.63 | 64.00 | 83.75 | 35.01 | 5.00 | 9.50 | 58.72 | 52.17 |
| Llama2-13B | StreamingLLM | 12.89 | 14.57 | 26.60 | 10.71 | 10.68 | 4.05 | 22.06 | 19.62 | 23.88 | 68.50 | 84.81 | 45.60 | 3.50 | 15.75 | 44.09 | 43.89 |
| | H2O | 13.43 | 15.11 | 27.14 | 11.25 | 11.22 | 4.59 | 22.60 | 20.16 | 24.42 | 69.00 | 85.35 | 46.14 | 5.00 | 16.25 | 44.63 | 44.43 |
| | HeadKV | 14.00 | 16.05 | 27.53 | 11.15 | 12.32 | 4.10 | 22.89 | 20.52 | 24.63 | 69.50 | 85.36 | 47.19 | 5.00 | 15.75 | 45.10 | 44.10 |
| | SnapKV | 13.53 | 15.49 | 26.77 | 11.74 | 10.86 | 5.23 | 23.58 | 20.47 | 25.14 | 69.50 | 85.68 | 46.44 | 5.00 | 17.25 | 45.10 | 46.04 |
| | PyramidKV | 13.95 | 14.99 | 28.32 | 12.05 | 11.69 | 5.63 | 22.52 | 20.68 | 24.70 | 69.50 | 86.21 | 45.98 | 5.00 | 17.75 | 45.10 | 44.36 |
| | CAKE | 13.71 | 15.39 | 27.42 | 11.53 | 11.50 | 4.87 | 22.88 | 20.44 | 24.70 | 69.38 | 85.63 | 46.42 | 5.50 | 16.80 | 44.91 | 44.71 |
| | SCORE(ours) | 13.49 | 16.45 | 27.91 | 12.45 | 12.49 | 4.71 | 23.90 | 20.61 | 25.66 | 68.50 | 87.75 | 47.73 | 5.00 | 17.75 | 45.67 | 46.97 |
| Llama3-8B-Instruct | StreamingLLM | 24.54 | 33.31 | 41.46 | 43.95 | 32.30 | 18.66 | 22.20 | 21.90 | 23.44 | 71.00 | 89.60 | 40.07 | 5.00 | 69.25 | 56.45 | 53.75 |
| | H2O | 25.08 | 33.85 | 42.00 | 44.49 | 32.84 | 19.20 | 22.74 | 22.44 | 23.98 | 71.50 | 90.14 | 40.61 | 5.50 | 69.50 | 57.00 | 54.29 |
| | HeadKV | 25.77 | 34.67 | 43.11 | 45.02 | 34.04 | 20.28 | 22.95 | 23.02 | 24.33 | 73.00 | 90.56 | 41.63 | 6.00 | 69.50 | 57.96 | 55.59 |
| | SnapKV | 25.61 | 33.63 | 43.28 | 44.78 | 33.78 | 20.58 | 23.07 | 22.62 | 24.31 | 71.50 | 90.44 | 40.66 | 6.00 | 68.50 | 57.33 | 55.30 |
| | PyramidKV | 25.06 | 34.44 | 40.82 | 44.87 | 31.91 | 17.93 | 23.41 | 22.89 | 24.49 | 72.00 | 90.61 | 40.74 | 6.00 | 69.50 | 57.24 | 53.19 |
| | CAKE | 25.38 | 34.15 | 42.30 | 44.79 | 33.14 | 19.50 | 23.04 | 22.74 | 24.28 | 72.07 | 90.44 | 40.91 | 5.50 | 69.50 | 57.30 | 54.59 |
| | SCORE(ours) | 25.68 | 38.26 | 44.27 | 45.81 | 36.53 | 21.05 | 24.50 | 23.13 | 25.51 | 74.00 | 90.64 | 41.63 | 5.00 | 68.50 | 57.52 | 53.62 |
| Mistral-7B-Instruction-v0.3 | StreamingLLM | 26.72 | 33.71 | 49.98 | 44.55 | 31.82 | 22.40 | 22.10 | 19.27 | 20.75 | 68.50 | 84.74 | 41.75 | 4.00 | 80.00 | 47.47 | 48.57 |
| | H2O | 27.26 | 34.25 | 50.52 | 45.09 | 32.36 | 22.94 | 22.64 | 19.81 | 21.29 | 69.00 | 85.28 | 42.29 | 4.60 | 80.50 | 48.01 | 49.11 |
| | HeadKV | 27.97 | 35.21 | 51.28 | 45.64 | 32.90 | 24.37 | 22.94 | 20.94 | 21.65 | 70.50 | 85.41 | 42.80 | 5.00 | 82.00 | 48.97 | 50.49 |
| | SnapKV | 28.11 | 34.54 | 50.70 | 45.38 | 32.92 | 22.70 | 23.36 | 19.90 | 21.73 | 70.00 | 85.50 | 43.07 | 5.00 | 80.50 | 48.58 | 49.62 |
| | PyramidKV | 26.89 | 34.21 | 50.78 | 45.46 | 32.46 | 22.96 | 22.81 | 19.80 | 21.73 | 68.50 | 86.13 | 42.20 | 5.00 | 81.00 | 47.67 | 48.43 |
| | CAKE | 26.87 | 35.09 | 50.21 | 45.06 | 32.38 | 23.09 | 23.05 | 19.82 | 21.56 | 68.93 | 85.55 | 42.39 | 4.50 | 81.10 | 48.06 | 49.13 |
| | SCORE(ours) | 26.82 | 37.71 | 50.35 | 45.53 | 32.97 | 24.82 | 24.18 | 20.97 | 22.45 | 69.50 | 86.21 | 43.09 | 5.50 | 83.00 | 49.13 | 50.55 |

Table 11: Performance comparison over 16 datasets from LongBench. Results are measured with a cache size of 1024. The best score is marked in **bold**, and the second best is marked with underline.

| Model | Method | Single-Document QA | | | Multi-Document QA | | | Summarization | | | Few-shot Learning | | | Synthetic | | Code | |
|---|---|---|---|---|---|---|---|---|---|---|---|---|---|---|---|---|---|
| | | NtvQA | Qasper | MF-en | HotpotQA | 2WikiMQA | Musique | GovReport | QMSum | MultiNews | TREC | TriviaQA | SAMSum | PCount | PR-en | Lcc | RB-P |
| Llama2-7B | StreamingLLM | 15.28 | 18.90 | 31.55 | 28.54 | 22.38 | 7.00 | 19.22 | 17.12 | 22.31 | 61.00 | 81.52 | 37.84 | 3.00 | 3.00 | 46.66 | 47.94 |
| | H2O | 17.01 | 20.74 | 32.04 | 30.37 | 24.64 | 8.90 | 20.59 | 18.88 | 24.43 | 63.00 | 81.95 | 39.98 | 5.50 | 4.00 | 45.69 | 48.79 |
| | HeadKV | 18.99 | 21.04 | 35.40 | 31.29 | 25.26 | 10.63 | 21.60 | 20.64 | 26.39 | 64.00 | 83.55 | 40.57 | 6.00 | 7.00 | 58.42 | 52.60 |
| | SnapKV | 17.79 | 21.91 | 35.68 | 31.96 | 26.21 | 9.61 | 22.1 | 21.08 | 25.01 | 64.00 | 82.95 | 40.84 | 6.00 | 7.50 | 57.78 | 48.54 |
| | PyramidKV | 17.25 | 21.02 | 37.33 | 31.29 | 25.50 | 9.80 | 22.45 | 20.61 | 24.97 | 64.00 | 83.81 | 40.09 | 6.00 | 8.00 | 56.98 | 51.78 |
| | CAKE | 18.44 | 21.95 | 35.95 | 31.60 | 25.01 | 10.10 | 22.75 | 20.26 | 25.09 | 64.00 | 85.99 | 40.59 | 6.00 | 7.00 | 52.52 | 51.98 |
| | SCORE(ours) | 18.47 | 22.04 | 35.99 | 31.64 | 25.58 | 10.51 | 23.75 | 20.58 | 25.39 | 64.00 | 89.39 | 41.00 | 6.00 | 7.50 | 58.58 | 52.87 |
| Llama2-13B | StreamingLLM | 10.40 | 11.80 | 23.70 | 8.93 | 9.34 | 1.31 | 22.54 | 17.51 | 23.36 | 65.00 | 57.55 | 44.99 | 2.00 | 14.75 | 39.01 | 40.07 |
| | H2O | 12.15 | 12.12 | 25.10 | 10.66 | 10.33 | 3.34 | 24.39 | 19.35 | 25.61 | 66.50 | 86.95 | 46.88 | 3.50 | 16.75 | 40.23 | 41.02 |
| | HeadKV | 13.80 | 16.46 | 28.42 | 12.00 | 13.21 | 4.59 | 23.63 | 20.97 | 25.76 | 69.00 | 86.83 | 41.89 | 4.00 | 14.75 | 45.83 | 46.47 |
| | SnapKV | 13.38 | 15.69 | 28.36 | 10.68 | 13.33 | 5.05 | 24.71 | 20.76 | 25.83 | 69.50 | 85.84 | 41.41 | 3.55 | 15.75 | 45.83 | 46.93 |
| | PyramidKV | 13.31 | 16.34 | 28.10 | 11.92 | 12.36 | 6.50 | 24.37 | 20.47 | 26.00 | 71.50 | 87.34 | 42.24 | 3.54 | 16.75 | 45.56 | 45.34 |
| | CAKE | 13.55 | 16.37 | 27.20 | 12.10 | 13.24 | 4.18 | 25.58 | 20.57 | 26.00 | 68.50 | 87.05 | 47.99 | 4.50 | 15.75 | 42.68 | 44.01 |
| | SCORE(ours) | 14.14 | 16.56 | 27.07 | 12.31 | 12.50 | 5.01 | 25.89 | 20.75 | 26.36 | 68.50 | 87.75 | 42.32 | 4.50 | 16.75 | 45.70 | 47.19 |
| Llama3-8B-Instruct | StreamingLLM | 22.84 | 34.94 | 40.61 | 43.54 | 33.06 | 19.86 | 24.24 | 20.30 | 21.36 | 69.00 | 80.91 | 40.09 | 3.55 | 67.50 | 50.26 | 48.20 |
| | H2O | 23.49 | 35.44 | 40.01 | 44.67 | 34.52 | 21.13 | 24.79 | 21.04 | 21.59 | 69.00 | 90.10 | 40.96 | 5.00 | 69.00 | 50.16 | 49.32 |
| | HeadKV | 25.79 | 37.69 | 43.56 | 45.72 | 36.23 | 20.25 | 24.38 | 22.93 | 25.72 | 74.00 | 90.56 | 41.53 | 5.39 | 69.25 | 57.77 | 54.34 |
| | SnapKV | 25.76 | 36.41 | 43.38 | 45.16 | 34.29 | 20.40 | 24.65 | 22.90 | 25.58 | 73.00 | 90.56 | 41.23 | 5.39 | 69.25 | 57.16 | 54.75 |
| | PyramidKV | 25.56 | 36.39 | 42.54 | 45.55 | 34.61 | 22.05 | 21.96 | 22.74 | 25.68 | 72.50 | 90.56 | 41.44 | 5.75 | 69.25 | 57.01 | 54.66 |
| | CAKE | 25.09 | 37.34 | 44.11 | 45.30 | 34.49 | 21.49 | 26.59 | 22.45 | 24.03 | 72.50 | 90.61 | 42.11 | 5.00 | 69.15 | 53.25 | 49.97 |
| | SCORE(ours) | 25.41 | 39.32 | 45.03 | 45.38 | 37.06 | 22.30 | 26.78 | 23.40 | 26.41 | 74.00 | 90.64 | 42.13 | 5.65 | 69.25 | 57.41 | 54.73 |
| Mistral-7B-Instruction-v0.3 | StreamingLLM | 24.93 | 32.81 | 46.42 | 43.67 | 28.77 | 22.05 | 21.99 | 19.09 | 18.29 | 67.00 | 83.64 | 40.34 | 3.50 | 80.00 | 43.15 | 46.39 |
| | H2O | 25.68 | 32.38 | 48.01 | 44.24 | 29.31 | 23.97 | 24.36 | 20.09 | 23.55 | 67.50 | 84.37 | 40.90 | 4.50 | 81.00 | 48.02 | 47.24 |
| | HeadKV | 27.76 | 36.91 | 50.13 | 45.80 | 32.83 | 24.36 | 24.61 | 21.09 | 23.15 | 70.50 | 85.33 | 43.44 | 5.50 | 83.00 | 49.26 | 50.06 |
| | SnapKV | 26.56 | 35.87 | 49.52 | 45.03 | 33.06 | 24.05 | 25.60 | 21.08 | 23.65 | 71.00 | 85.21 | 42.43 | 5.50 | 82.50 | 49.56 | 48.48 |
| | PyramidKV | 27.11 | 36.18 | 50.96 | 46.16 | 32.95 | 24.55 | 24.57 | 20.65 | 23.51 | 70.00 | 86.23 | 43.32 | 7.00 | 83.00 | 49.74 | 49.72 |
| | CAKE | 27.11 | 37.74 | 48.73 | 46.16 | 32.92 | 24.03 | 26.52 | 20.98 | 24.40 | 70.00 | 85.81 | 42.71 | 5.00 | 82.00 | 49.18 | 48.44 |
| | SCORE(ours) | 28.89 | 37.78 | 49.62 | 45.29 | 32.99 | 24.88 | 26.89 | 21.10 | 24.50 | 71.00 | 86.33 | 43.53 | 7.00 | 83.00 | 49.78 | 50.51 |

# F    LIMITATIONS

Despite its effectiveness in managing multi-level redundancy in the KV cache, *SCORE* has several limitations. (1) It requires sampling representations from each attention head and computing pairwise distances, which can be more computationally expensive than simple statistical metrics (*e.g.*, variance or entropy), especially as the number of samples increases. Although this trade-off does not significantly impact overall performance, it remains a consideration for large-scale or real-time applications. Our cascading design and sampling limits help reduce this cost, but further optimization may be needed for broader scalability. (2) *SCORE* relies on fixed distance metrics that may not perfectly align with downstream task performance. Integrating task-aware or learned similarity measures could improve redundancy estimation. (3) This work focuses on inference-time cache management; extending *SCORE* to training scenarios such as fine-tuning or continual learning remains an open direction for future research.

