# OpenReview forum: "SCORE: Similarity-Aware Contextual Overlap-Redundancy Eviction for Efficient KV Cache Compression in LLMs"
_ICLR.cc/2026/Conference — ICLR 2026 Conference Withdrawn Submission_

### Official Review · Reviewer_DDPd · 2025-10-14

**Soundness:** 2
**Presentation:** 2
**Contribution:** 2
**Rating:** 2
**Confidence:** 5

**Summary:**

This paper proposes SCORE (Similarity-aware Contextual Overlap-Redundancy Eviction), a novel and intricate framework for KV cache compression in LLMs. The method's core idea is to quantify structural redundancy within the KV cache by introducing a set of new metrics: Intra-layer Redundancy (IR), Temporal Deviation (TD), and Head-level Distinctness (HD). These metrics are then used in a hierarchical fashion to first allocate cache budgets dynamically across layers and heads, and finally to guide a greedy token selection process that penalizes redundant choices. The authors evaluate their method on long-context benchmarks like LongBench, showing that it outperforms existing state-of-the-art baselines, especially under tight memory constraints.

**Strengths:**

The paper addresses the critical problem of KV cache memory consumption for long-context inference, which is a major bottleneck for deploying large models.

The core idea of identifying and eliminating structural redundancy across layers and heads is a novel and more principled approach compared to methods that rely solely on attention scores or simpler heuristics.

The proposed multi-level metrics (IR, TD, HD) provide a fine-grained methodology for analyzing the internal representations of the model, which is a valuable contribution in itself.

**Weaknesses:**

* **Methodological Complexity and Lack of Sensitivity Analysis**: A significant weakness of this work is the complexity of the proposed framework. SCORE introduces a considerable number of new metrics and associated hyperparameters (e.g., `λ1`, `λ2` for novelty scoring, `γ` for the EMA update, `α` for the penalty term in token selection). This complexity makes the method difficult to implement and tune. Crucially, the paper **lacks a thorough ablation study on the sensitivity of these hyperparameters**. It is unclear how robust the method is to different parameter choices, which poses a significant challenge for practical deployment and reproducibility, as practitioners would have little guidance on how to set these values for new models or tasks.

* **Marginal Performance Improvement**: While SCORE often appears as the top-performing method, the reported improvements over strong, recent baselines like CAKE and HeadKV are frequently **very marginal**. For instance, in many of the sub-tasks presented in Table 1, the performance gain is less than a single percentage point. This raises questions about the practical significance of the results and whether the added complexity of the SCORE framework is justified by such a small performance lift.

* **Insufficient Experimental Scope and Clarity**: The experimental evaluation, while broad in task coverage, has limitations. **Table 1 only presents results for a single, extremely compressed cache size (128)**, which does not provide a full picture of how the methods compare at various compression ratios. While Figure 4 shows trends, a detailed table for other cache sizes (e.g., 256, 512) would be necessary for a complete comparison. Furthermore, **Table 1 lacks an overall average score** across the tasks, forcing the reader to manually assess the results and making it difficult to draw a clear, aggregate conclusion.

* **Poor Presentation of Figures**: The visual presentation of the results needs significant improvement. In several key figures (e.g., Figure 4), the color palette used for different methods consists of very similar shades of blue, green, and purple. This makes the lines and points **very difficult to distinguish**, undermining the clarity of the comparisons.

**Questions:**

1.  Given the introduction of multiple new hyperparameters, could the authors provide a sensitivity analysis for `λ1`, `λ2`, `γ`, and `α`? How much does performance vary with changes to these parameters, and is there a principled way to set them without extensive grid searching for each new model or dataset?
2.  The performance gains over the next-best methods in Table 1 are often quite small. Could the authors comment on the complexity-versus-performance trade-off? In which specific scenarios or task types do they believe the sophisticated redundancy modeling of SCORE provides a clear and significant advantage that justifies its complexity over simpler methods like CAKE or HeadKV?
3.  For a more comprehensive evaluation, would it be possible to provide an average score in Table 1? Additionally, could the authors provide detailed tables (similar to Table 1) for other cache sizes to better illustrate how the performance gap between SCORE and the baselines evolves with different memory budgets?
4.  Would the authors consider revising the color schemes in the plots to use more distinct and easily distinguishable colors to improve the readability of the experimental results?

---

### Official Review · Reviewer_q3YE · 2025-10-25

**Soundness:** 2
**Presentation:** 3
**Contribution:** 1
**Rating:** 2
**Confidence:** 5

**Summary:**

This paper addresses the memory inefficiency problem in LLMs inference caused by the KV cache. It proposes SCORE to reduce KV cache redundancy by introducing a distance-based multi-level similarity metric. The method quantifies representational redundancy across both layers and heads, reallocates cache budgets dynamically, and selects informative tokens through a redundancy-aware greedy selection. The approach enables finer-grained, context-sensitive cache compression. Experimental results on LongBench, LongBench-v2, NeedleBench, and InfiniteBench show that SCORE retains about 95% of full-cache performance using only 1.5% of memory, consistently outperforming prior methods such as CAKE, SnapKV, and HeadKV under strict memory constraints.

**Strengths:**

1. The paper tackles an important and practical problem. KV cache efficiency in long-context LLM inference, which is central to scaling LLMs.
2. The motivation is clear and well-grounded, supported by analysis of redundancy at multiple levels (layer, head, token).
3. Experimental results are comprehensive and convincing, covering multiple benchmarks, models (LLaMA2, LLaMA3, Mistral), and settings, demonstrating strong robustness and generality.

**Weaknesses:**

1. Although effective, the proposed method is incremental over existing eviction-based works such as CAKE and HeadKV, mainly extending them with redundancy-aware similarity metrics rather than introducing a fundamentally new paradigm.

2. The evaluation lacks real-world latency–accuracy tradeoff benchmarks on end-to-end generative tasks (e.g., AIME or instruction-following), making it hard to assess SCORE’s deployment impact.

**Questions:**

same in weakness

---

### Official Review · Reviewer_3fgP · 2025-10-30

**Soundness:** 3
**Presentation:** 3
**Contribution:** 3
**Rating:** 4
**Confidence:** 4

**Summary:**

The paper proposes SCORE (Similarity-aware Contextual Overlap-Redundancy Eviction), a novel framework designed for efficient KV (Key-Value) cache compression in large language models (LLMs) during long-context inference. The KV cache is a critical memory bottleneck, and SCORE addresses this by eliminating structural redundancy.
The paper's main contributions are:
·Distance-Based Redundancy Metric: SCORE introduces the first distance-based metric to precisely measure and eliminate redundancy within the KV cache. This is an improvement over prior work that relies on heuristic or single-axis importance metrics.
·Redundancy-Aware Multi-level Metric: It uses multi-level similarity metrics (Intra-layer Redundancy, Temporal Deviation, and Head-level Distinctness) to quantify redundancy across both the layers and attention heads of the transformer model. This captures hierarchical information flow and representational diversity, jointly addressing redundancy across layer-head interactions.
·Hierarchical Budget Allocation: The framework dynamically reallocates cache budgets across layers and heads based on these redundancy scores, which prioritizes informative and non-redundant tokens under strict memory constraints.
·Greedy Token Selection: A redundancy-aware greedy token selection algorithm is used to maximize information diversity in the cache by penalizing tokens that have already been redundantly selected by multiple heads.
·Performance: Extensive experiments on long-context benchmarks show that SCORE retains 95% of the full KV cache performance while using only 1.5% of the cache, consistently outperforming state-of-the-art baselines.

**Strengths:**

Originality: The paper introduces a highly original perspective by defining KV cache redundancy using a distance-based metric for elimination, moving beyond the heuristic or single-axis importance scores of prior work. The core innovation is the Similarity-aware Contextual Overlap-Redundancy Eviction (SCORE) framework, which creatively combines multi-level, hierarchical similarity metrics (Intra-layer Redundancy, Temporal Deviation, and Head-level Distinctness). This combination allows for the first time an intricate measurement and management of redundancy across the layer and attention head dimensions simultaneously, tackling the problem with an unprecedented degree of sophistication.
·Quality: The overall quality is excellent, marked by a sound technical foundation and robust empirical validation. The proposed multi-level metrics are well-defined and mathematically grounded in similarity/distance measures. The resulting hierarchical budget allocation and redundancy-aware greedy selection algorithm form a coherent and complete system. The paper demonstrates high-quality results, achieving 95% of full KV cache performance while using a drastically reduced cache size of only 1.5%, which is a highly significant and convincing efficiency gain.
·Clarity: The paper is clear and well-structured. It logically breaks down the problem, introduces the novel distance-based redundancy concept, and then details the three component similarity metrics. The overall framework, including the hierarchical budget allocation and greedy selection, is presented in an easy-to-follow manner. Key concepts are defined precisely, allowing readers to understand the technical details of the SCORE mechanism. The figures and experimental sections are well-organized and effectively communicate the system's effectiveness and its superiority over baselines.
·Significance: The paper is highly significant for the field of efficient LLM inference. The KV cache is a critical memory bottleneck for long-context LLMs, directly limiting the scalability and deployment of these models. By effectively retaining near-full performance with a ~66x reduction in cache size (1.5% usage), SCORE offers a practical and substantial solution to a major architectural challenge. Its similarity-aware approach provides a new and fundamental direction for KV cache compression research, establishing a new state-of-the-art that will likely serve as a foundational benchmark for future work in this area.

**Weaknesses:**

1. Complexity and Computational Overhead of Redundancy Metrics
The paper convincingly demonstrates memory savings (cache size) and performance (accuracy), but a detailed runtime or throughput analysis of the overhead introduced by calculating the three complex similarity metrics (Intra-layer Redundancy, Temporal Deviation, Head-level Distinctness) and performing the redundancy-aware greedy selection is often missing or could be expanded. This overhead could potentially negate some of the inference speedup gained from reduced memory bandwidth.
·Actionable Insight: The authors should provide a breakdown of the inference time or tokens-per-second throughput comparing the full KV cache, the baseline methods, and SCORE, specifically separating the time spent on attention computation from the time spent on the SCORE eviction logic. Furthermore, explore and demonstrate a less complex, approximate scoring mechanism (e.g., using a small, pre-trained predictor network or a simpler metric) that trades off a small amount of accuracy for a significant reduction in the scoring overhead.

2. Generalizability Across Diverse Model Architectures and Tasks
The generalizability of SCORE's core mechanism—which relies on specific interactions within the standard Multi-Head Attention (MHA) block—to models with fundamentally different memory mechanisms is not fully explored. For instance, its applicability to Mixture-of-Experts (MoE) models, or to other memory-augmented architectures, remains an open question.
·Actionable Insight: Supplement the current experiments by applying SCORE to a prominent MoE model (e.g., Mixtral) to assess if the Hierarchical Budget Allocation needs modification to account for sparse expert activation. Additionally, demonstrating the framework's effectiveness on a non-LLM transformer task (e.g., a long-range vision task using a Vision Transformer) could broaden the perceived utility of the proposed redundancy-eviction principles.

3. Sensitivity Analysis of Hyperparameters
The paper does not provide a comprehensive sensitivity analysis to show how robust the method is to changes in these critical parameters. Overly sensitive hyperparameters would make the method brittle and difficult to deploy consistently in varied production environments.
·Actionable Insight: Include a visualization (e.g., a heatmap or line graph) illustrating the performance change (e.g., loss/perplexity) as the overall cache budget (B) and the weights for the multi-level metrics are varied. This would establish confidence in the method's stability and provide guidance for practitioners on selecting optimal configurations.

**Questions:**

1. Computational Overhead and Latency Analysis
·Question: The paper demonstrates exceptional memory savings and accuracy retention, but a critical factor for an inference technique is its real-world latency. Can the authors provide a detailed breakdown of the inference time (tokens/second throughput)? Specifically, how does the time spent on the SCORE eviction logic (calculating the three multi-level similarity metrics and performing the greedy selection) compare to the time saved from reduced memory bandwidth usage?
·Suggestion: Present a figure that clearly plots the total end-to-end latency versus the context length for the full KV cache, a leading baseline (e.g., H2O), and SCORE, along with a secondary axis or bar chart that isolates the percentage of total inference time consumed by the eviction mechanism for each method. This would establish whether the method is truly beneficial for low-latency deployment, not just memory-constrained scenarios.

2. Sensitivity of Multi-level Similarity Metrics
·Question: The efficacy of SCORE relies on the combination of the three metrics: Intra-layer Redundancy, Temporal Deviation, and Head-level Distinctness. Were these metrics simply averaged, or was a weighted combination used? If a weighted combination was used, how were these weights determined (e.g., manually tuned, learned via meta-learning, or fixed)? Furthermore, how sensitive is the final performance to changes in the relative weighting of these three components?
·Suggestion: Include an ablation study that systematically removes one or two of the similarity metrics (e.g., using only Intra-layer Redundancy, or only Temporal Deviation) to quantify the unique contribution of each component to the final performance gain. This would rigorously validate the necessity of the multi-level design over a simpler, single-metric approach.

3. Eviction Frequency and Token Age
·Question: Is the SCORE eviction process (re-evaluating all tokens, calculating scores, and performing greedy selection) executed at every single new token generation step, or only after a certain buffer of tokens is accumulated, or perhaps when the cache hits a predefined usage threshold? If it's executed at every step, the overhead is amplified.
·Suggestion: Discuss the potential for integrating a concept of token age or recency into the SCORE mechanism. Could tokens that have already survived several eviction cycles be given a small "score bonus" to reduce the likelihood of costly re-calculation and re-eviction, thus introducing a bias towards older, stable context tokens? This could be a way to further optimize the computation.

4. Generalizability to Alternative Architectures
·Question: The framework is validated primarily on standard decoder-only transformers (LLaMA/Mistral). How would the SCORE mechanism need to be adapted, if at all, to be applied to transformer models with different attention or memory patterns, such as Multi-Query Attention (MQA), Grouped-Query Attention (GQA), or Mixture-of-Experts (MoE) models where the head redundancy might be less pronounced or the layer interactions are fundamentally different?
·Suggestion: Even without full experimental results, a conceptual discussion on the anticipated challenges and necessary modifications for deploying SCORE on GQA or MoE models would significantly increase the paper's perceived scope and provide a clear roadmap for future work.

---

### Official Review · Reviewer_7pT2 · 2025-11-03

**Soundness:** 3
**Presentation:** 3
**Contribution:** 2
**Rating:** 4
**Confidence:** 3

**Summary:**

This paper addresses the memory bottleneck caused by the KV cache in LLMs during long-context inference. Existing cache eviction methods often depend on simple heuristics and fail to account for structural redundancy across layers and attention heads. The paper introduces SCORE, a framework that uses a distance-based similarity metric to quantify and remove redundant information. Experimental results on long-context benchmarks like LongBench show that SCORE maintains 95% of full KV cache performance using as little as 1.5% of the cache.

**Strengths:**

1. The paper introduces a novel distance-based similarity metric to directly quantify and manage structural redundancy in the KV cache, moving beyond standard attention-based heuristics (Section 3.2).
2. The evaluation is extensive, testing on multiple models, cache sizes, and diverse long-context benchmarks like LongBench V2 and InfiniteBench, demonstrating the method's generalizability (Section 4, Appendices).

**Weaknesses:**

### About Method

1.  The head-wise budget allocation strategy (Equation 12) multiplies head distinctness (HD) with contribution (T). This design may lead to excessive budget concentration on a few "star" heads while neglecting other useful ones. The paper should provide a more thorough justification for the superiority of this multiplicative model or compare it against alternatives, such as an additive model.
2.  In the greedy token selection mechanism (Equation 13), calculating the penalty requires tracking the selection count `r_s` for each token across all heads. The paper claims this mechanism incurs "little additional overhead" but lacks a concrete analysis of the time and space complexity required to implement this counter, making the claim unconvincing.
3.  The methodology heavily relies on the "cascading strategy" from the prior work CAKE, yet fails to briefly introduce its basic mechanism in the main text. This requires readers to have prior knowledge and reduces the paper's self-containedness and readability.

### About Experiment

1.  The most critical deficiency in the experimental section is the lack of ablation studies on the core components. The authors should add experiments to independently validate the effectiveness of the hierarchical budget allocation, the head-level allocation, and the redundancy-aware greedy selection penalty, to demonstrate the actual contribution of each part of the method.
2.  The authors need to acknowledge and analyze the method's performance limitations in certain scenarios. For instance, on several tasks in LongBench (Table 1), SCORE's performance is significantly lower than baselines like HeadKV, yet the paper provides no explanation for this, which undermines the rigor of the study.
3.  The experimental evaluation is missing a crucial and standard metric: PPL. It is recommended to supplement the evaluation with PPL on a standard language modeling dataset (e.g., PG-19) to more fundamentally measure how well the method preserves the model's predictive capabilities.
4.  The core claim of "preserving semantic diversity" lacks experimental evidence. It is suggested to add qualitative (e.g., visualizing the semantic distribution of kept tokens) or quantitative (e.g., introducing a semantic diversity metric) analysis to substantiate this claim.

**Questions:**

1.  Regarding the head budget allocation in Equation (12), could you elaborate on why you chose to multiply head distinctness (`HD_{l,h}`) and contribution (`T_{l,h}`) rather than using an additive approach like a weighted sum? Was this multiplicative design experimentally validated against other aggregation methods to confirm its robustness in preventing excessive budget concentration?
2.  For the greedy token selection in Equation (13), could you provide the specific implementation details and a complexity analysis for tracking the cross-head selection count `r_s`? How is the claim of "little additional overhead" substantiated, especially when processing sequences with millions of tokens?

---

### Note · Authors · 2025-11-19

I have read and agree with the venue's withdrawal policy on behalf of myself and my co-authors.